# Multi-agent Coordination via Flow Matching

**Dongsu Lee**[1]    **Daehee Lee**[2]    **Amy Zhang**[1]
[1]The University of Texas at Austin    [2]Sungkyunkwan University
dongsu.lee@utexas.edu

## Abstract

This work presents `MAC-Flow`, a simple yet expressive framework for multi-agent coordination. We argue that requirements of effective coordination are twofold: *(i)* a rich representation of the diverse joint behaviors present in offline data and *(ii)* the ability to act efficiently in real time. However, prior approaches often sacrifice one for the other, *i.e.*, denoising diffusion-based solutions capture complex coordination but are computationally slow, while Gaussian policy-based solutions are fast but brittle in handling multi-agent interaction. `MAC-Flow` addresses this trade-off by first learning a flow-based representation of joint behaviors, and then distilling it into decentralized one-step policies that preserve coordination while enabling fast execution. Across four different benchmarks, *including* 12 environments and 34 datasets, `MAC-Flow` alleviates the trade-off between performance and computational cost, specifically achieving about ×**14.5** faster inference compared to diffusion-based MARL methods, while maintaining good performance. At the same time, its inference speed is similar to that of the prior Gaussian policy.

**Code**: https://github.com/DongsuLeeTech/mac-flow

TL;DR: **MAC-Flow alleviates performance-inference time trade-off**, achieving a **14.5x speedup** compared to SOTA

**Performance. vs. Inference time.** (Benchmarks: SMACv1 and SMACv2)

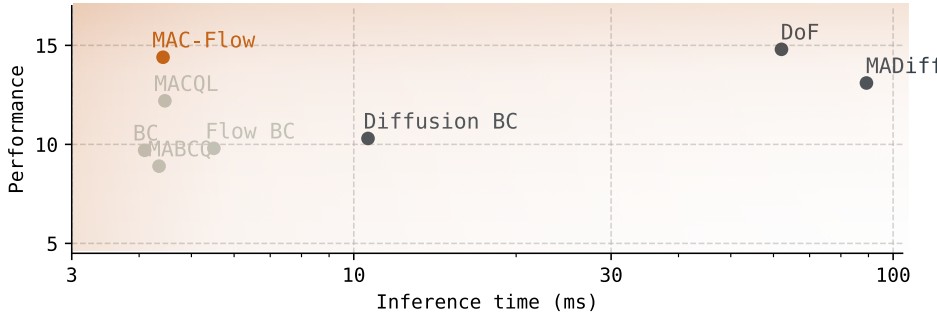

Figure 1: **Summary of results.** This summarizes *performance* vs. *inference speed* for selected algorithms on widely-used MARL benchmarks, SMACv1 and SMACv2. We plot aggregate mean performance and inference time across 18 datasets for 8 scenarios related to the SMAC maps. More precisely, we measure inference time based on the total computation performed by each algorithm and report it by using milliseconds (`ms`) unit and log scale, where a higher value indicates greater computational cost. As a result, our proposed solution, `MAC-Flow`, achieves ×14.5 faster inference speed on average with comparable performance compared to previous SOTA.

## 1 Introduction

Multi-agent reinforcement learning (MARL) has been actively studied to handle real-world problems in multi-agent systems, such as modular robot control (Zhang et al., 2024; Peng et al., 2021), multi-player strategy games (Carroll et al., 2019; Samvelyan et al., 2019), and autonomous driving (Lee et al., 2024a; Vinitsky et al., 2022). However, because seminal MARL approaches heavily rely on extensive online interactions among agents, their applicability to real-world domains is severely limited. Moreover, training from scratch is both expensive and risky in data collection and computation (Levine

et al., 2020). These challenges motivate methods that import the safety and efficiency benefits of offline reinforcement learning (RL) into MARL (Wang et al., 2021).

Offline RL learns decision-making policies from fixed datasets, thereby avoiding expensive and risky environment interactions by optimizing returns while staying close to the dataset's state-action distribution (Bhargava et al., 2024; Kumar et al., 2020; Wu et al., 2019). However, the transition from a single-agent to a multi-agent system is not trivial, introducing a unique set of formidable challenges. For example, the joint action space grows exponentially with the number of agents, making it difficult to learn coordinated behaviors that adapt to diverse interaction patterns (Li et al., 2025c; Liu et al., 2025; Barde et al., 2024). Despite several efforts to address these problems (Yang et al., 2021; Pan et al., 2022; Wang et al., 2023), they still struggle with the complex multi-modality of joint action distributions, as Gaussian policies are prone to failure by generating out-of-distribution coordination.

To tackle this challenge, seminal works have turned to powerful generative models based on denoising diffusion in MARL (Zhu et al., 2024; Li et al., 2025a). While diffusion policies capture complex multi-modal distributions, directly optimizing them with respect to a learned value function remains non-trivial. Backpropagating value gradients through a multi-step denoising chain is expensive and unstable (Park et al., 2025; Ma et al., 2025; Yang et al., 2025), so several previous RL solutions employ diffusion for offline imitation or a distillation-based solution (Ding & Jin, 2023; Chen et al., 2024b). Furthermore, their reliance on an iterative denoising process, which requires numerous neural function evaluations per forward pass, makes them computationally expensive at inference time. This latency precludes its practicality in scenarios that require time-critical decision-making.

At its core, the question motivating this study can be phrased as follows:

> *How should offline MARL algorithms balance the trade-off*
> *between expressiveness for multi-agent distributions and computational efficiency?*

Building on this question, this work introduces a novel offline MARL framework, dubbed `MAC-Flow`, that bridges the gap between expressive generative policies (simply coordination performance) and computational demands. Our main idea is to first learn a flow-based joint policy through behavioral cloning, which captures the complex multi-modal distribution of joint actions in offline multi-agent datasets. For decentralized execution, we factorize the flow-based joint policy into a one-step sampling policy by optimizing two objectives: *(i)* distilling the flow-based joint action distribution and *(ii)* maximizing the global value function under the individual-global-max (IGM) principle (Son et al., 2019). By decoupling expressiveness from value maximization, `MAC-Flow` avoids the instability of backpropagation through time (Park et al., 2025; Wagenmaker et al., 2025).

Our contributions are summarized as follows.

- We propose `MAC-Flow`, a novel MARL algorithm that enables scalable multi-agent systems by alleviating the trade-off between coordination performance and inference speed.
- We introduce a flow-based joint policy factorization method that decouples expressiveness from optimization, decomposing a flow-based joint policy into efficient one-step sampling policies while jointly optimizing RL and imitation objectives with mathematical guarantees.
- We demonstrate the efficiency of `MAC-Flow` across four widely used MARL benchmarks, 12 environments, and 34 datasets, encompassing diverse characterizations, especially in $14.5\times$ speedup compared to previous diffusion-based solutions in SMAC benchmarks.

## 2 RELATED WORK

**Offline MARL.** The goal of offline RL is to extract a policy using static operational logs, without active interaction with the environment. In multi-agent settings, inter-agent dependencies amplify distribution shift. Even slight deviations in a single agent's policy can trigger cascading mismatches in joint behavioral patterns, complicating efforts to maximize returns while preserving alignment with the dataset's joint state-action distribution. Naive approaches typically involve extending single-agent offline RL methods to multi-agent contexts (Fujimoto et al., 2019; Kumar et al., 2020; Fujimoto & Gu, 2021). Yet, these extensions often fall short in incorporating global regularization to facilitate cooperation. More sophisticated methods, tailored for multi-agent settings, tackle coordination issues via value decomposition, policy factorization, and model-based optimization (Yang et al., 2021; Pan

et al., 2022; Wang et al., 2023; Barde et al., 2024; Liu et al., 2025). Despite these advances, such techniques struggle to capture the multi-modal distributions inherent in multi-agent offline datasets, resulting in imprecise credit assignment. To better capture joint action distribution, generative modeling has gained traction, for example, diffusion-based trajectory modeling (Zeng et al., 2025) (Li et al., 2025a; Zhu et al., 2024), diffusion-based policy (Qiao et al., 2025; Li et al., 2023), and transformer-based modeling (Wen et al., 2022; Tseng et al., 2022; Meng et al., 2023b). However, they frequently require substantial computational resources and may struggle to learn optimal patterns, owing to their reliance on behavioral cloning (BC) paradigms. This work introduces a novel MARL algorithm that increases efficiency and takes advantage of both RL and generative modeling.

**Diffusion and Flow Matching in RL.** Drawing on the robust capacity of iterative generative modeling frameworks, such as denoising diffusion processes and flow matching techniques, recent efforts have explored the diverse ways to employ them for enhancing RL policies. Examples of RL with an iterative generative model include world modeling (Rigter et al., 2023; Ding et al., 2024; Alonso et al., 2024), trajectory planning (Janner et al., 2022; Ajay et al., 2022; Zheng et al., 2023; Liang et al., 2023; Ni et al., 2023; Chen et al., 2024a; Lu et al., 2025), policy modeling (Alles et al., 2025; Zhang et al., 2025b; Chi et al., 2023), data augmentation (Lu et al., 2023; Huang et al., 2024; Lee et al., 2024b; Yang & Wang, 2025), policy steering (Wagenmaker et al., 2025), and exploration (Ren et al., 2024; Liu et al., 2024; Li et al., 2025b). Such iterative generative models show powerful performance, but their inference is prohibitively slow for real-world deployment.

Our approach extracts an expressive flow policy to capture the multi-modal distribution of mixed behavioral policies from offline multi-agent datasets. Algorithmically, this is motivated by flow distillation (Frans et al., 2024; Chen et al., 2025) and flow Q-learning (FQL) (Park et al., 2025), which distills a one-step policy with an RL objective to model complex action distributions via flow matching in single-agent RL. Instead, we leverage a flow matching-based joint policy that explicitly models the joint action distribution across agents and pioneering the integration of the IGM principle (Son et al., 2019) with flow matching in MARL, ensuring individual policies align with the global optimal joint policy for enhanced coordination and scalability.

## 3 BACKGROUND

**Problem Formulation.** This work posits the MARL problems under a decentralized partially observable Markov decision process (Dec-POMDP) (Bernstein et al., 2002) $\mathcal{M}$ defined by a tuple $(\mathcal{I}, \mathcal{S}, \mathcal{O}_i, \mathcal{A}_i, \mathcal{T}, \Omega_i, r_i, \gamma)$, where $\mathcal{I} = \{1, 2, \cdots, I\}$ denotes a set of agents. Here, $\mathcal{S}$ represents the global state space; $\mathcal{O}_i$ and $\mathcal{A}_i$ correspond to the observation and action spaces specific to agent $i$, respectively. The state transition dynamics are captured by $\mathcal{T}(s'|s, \mathbf{a}) : \mathcal{S} \times \mathcal{A}_1 \times \cdots \times \mathcal{A}_I \mapsto \mathcal{S}$, where $\mathbf{a}$ is a joint action $[a_1, a_2, \cdots, a_I]$ and we color the gray to denote placeholder variables. Next, $\Omega_i(o_i|s) : \mathcal{S} \mapsto \mathcal{O}_i$ specifies the observation function for agent $i$. Each agent $i$ receives individual rewards according to its reward function $r_i(s, a_i, \mathbf{a}_{-i}) : \mathcal{S} \times \mathcal{A}_1 \times \cdots \times \mathcal{A}_I \mapsto \mathbb{R}$. The goal of offline MARL under cooperative setups is to learn a set of policies $\Pi = \{\pi_1, \pi_2, \cdots, \pi_I\}$ that jointly maximize the discounted cumulative reward $\mathbb{E}_{\boldsymbol{\tau} \sim p^{\Pi}(\tau)} \left[ \sum_{i=1}^{I} \sum_{h=0}^{H} \gamma^h r_i(s^h, a_i^h, \mathbf{a}_{-i}^h) \right]$ from an offline multi-agent dataset $\mathcal{D} = \{\boldsymbol{\tau}^{(n)}\}_{n \in \{1, 2, \cdots, N\}}$ without environment interactions, where $\gamma \in [0, 1)$ is a discounted factor, $\boldsymbol{\tau}$ denotes a joint trajectory $\{\tau_1, \tau_2, \cdots, \tau_I\}$, $\tau_i = (o_i^0, a_i^0, \cdots, o_i^H, a_i^H)$, and $p^{\Pi}(\boldsymbol{\tau})$ represents probability distribution over joint trajectories induced by a set of policies $\Pi$.

**Individual-Global-Max Principle.** The IGM principle serves as a foundational approach in MARL, offering a method to ensure globally consistent action selection through factorized Q-value functions for each agent (Son et al., 2019; Rashid et al., 2020). By aligning individual agent policies with a shared objective, IGM simplifies the complexity of joint action spaces, making it a scalable solution for multi-agent systems. This principle can be mathematically defined as follows.

$$\arg\max_{\mathbf{a}} Q_{\text{tot}}(\boldsymbol{o}, \mathbf{a}) = \left( \arg\max_{a_1} Q_1(o_1, a_1), \ldots, \arg\max_{a_I} Q_I(o_I, a_I) \right) \tag{1}$$

Here, $Q_{\text{tot}}(\mathbf{o}, \mathbf{a})$ represents the global Q function, while $Q_i(o_i, a_i)$ denotes an individual Q function for agent $i$. The IGM ensures that optimizing its local $Q_i$ remains consistent with the global optimum.

**Behavioral-regularized Offline RL.** Behavioral regularization (Wu et al., 2019; Fujimoto & Gu, 2021; Kostrikov et al., 2021; Tarasov et al., 2023; Eom et al., 2024) is a simple and powerful way to

alleviate the out-of-distribution issue in the offline RL setting. Most seminal works leverage both actor and critic penalization, whereas critic penalization may deteriorate the Q function in additional online training. Therefore, to secure the versatility in both offline and online, we minimize the most vanilla loss functions of the behavioral-regularized actor-critic framework as follows.

$$\mathcal{L}_Q(\theta) = \mathbb{E}_{(o,a,o',r)\sim\mathcal{D},\ a_i'\sim\pi_\phi}\left[Q_\theta(o,a) - \left(r + \gamma Q_{\bar\theta}(o',a')\right)\right] \tag{2}$$

$$\mathcal{L}_\pi(\phi) = \mathbb{E}_{(o,a)\sim\mathcal{D},\ a^\pi\sim\pi_\phi}\left[-Q_\theta(o,a^\pi) + \alpha\underbrace{f\left(\pi_\phi(a|o),\mu(a|o)\right)}_{\text{behavioral regularization}}\right] \tag{3}$$

Herein, $\theta$ and $\phi$ is a parameter of critic and actor networks respectively, $\bar\theta$ is a target parameter of the critic network (Mnih et al., 2013), $\alpha$ is a weight coefficient (Fujimoto & Gu, 2021), $f(\cdot,\cdot)$ represents the function that captures the divergence between a trained policy $\pi_\phi(a|o)$ and offline policy $\mu(a|o)$, which is used to collect the dataset $\mathcal{D}$. The simplest implementation of $f(\cdot,\cdot)$ is the entropy regularization or behavioral cloning in the soft-actor critic algorithm $-\log\pi_\phi(a|o)$ (Haarnoja et al., 2018). Within such an RL framework, we introduce a flow-based MARL solution.

**Flow Matching and Flow Policies.** Flow matching (Lipman et al., 2022; Gat et al., 2024) offers an alternative to denoising diffusion models, which rely on stochastic differential equations (SDEs) (Ho et al., 2020; Song et al., 2020a; Nichol & Dhariwal, 2021). It simplifies training and speeds up inference while often maintaining quality, as it is based on ordinary differential equations (ODEs) (Papamakarios et al., 2021; Wildberger et al., 2023; Lipman et al., 2024).

The objective of flow matching is simple: to transform a simple noise distribution $p_0 = \mathcal{N}(0,\mathbf{I}_d)$ into a given target distribution $p_1 = p(x)$ over a $d$-dimensional Euclidean space $\mathcal{X} \subset \mathbb{R}^d$. More precisely, it finds the parameter $\phi$ of a time-dependant velocity field $v_\phi(t,x) : [0,1] \times \mathbb{R}^d \mapsto \mathbb{R}^d$ to build a time-dependent flow (Lee, 2003) $\psi_\phi(t,x) : [0,1] \times \mathbb{R}^d \mapsto \mathbb{R}^d$ via ODE as follows:

$$\frac{d}{dt}\psi_\phi(t,x) = v_\phi(t,\psi_\phi(t,x)), \quad \text{where} \quad \psi_\phi(0,x) = x.$$

Here, the terminal state $\psi_\theta(1,x^0)$, with $x^0 \sim p_0$, is expected to follow the target distribution $p_1$. To make the learning problem tractable, we follow an interpolating probability path $(p_t)_{0\le t\le 1}$ between $p_0$ and $p_1$, where each intermediate sample is obtained by linear interpolation as $x^t = (1-t)x^0 + tx^1$. A timestep $t$ is sampled from a uniform distribution $\text{Unif}([0,1])$ corresponding to a flow step.

The velocity field is then trained to approximate the displacement direction $(x^1 - x^0)$ at intermediate points along this path. Formally, the training objective is defined as:

$$\mathcal{L}(\phi) = \mathbb{E}_{x^0\sim p_0,\ x^1\sim p_1,\ t\sim\text{Unif}([0,1])}\left[\|v_\phi(t,x^t) - (x^1 - x^0)\|_2^2\right], \tag{4}$$

which encourages $v_\theta$ to recover the underlying transport field from source to target.

In this work, we extract a policy via the simplest variant of flow matching (Equation 4) (Park et al., 2025; Zhang et al., 2025a). Specifically, the flow policy is extracted to minimize the following loss.

$$\mathcal{L}_{\text{Flow-BC}}(\phi) = \mathbb{E}_{x^0\sim p_0,\ (o,a)\sim\mathcal{D},\ t\sim\text{Unif}([0,1])}\left[\|v_\phi(t,o,x^t) - (a - x^0)\|_2^2\right] \tag{5}$$

Flow policies extend flow matching to policy learning by conditioning the velocity field $v_\phi(t,o,x)$ on the observation $o$. The resulting flow $\psi_\phi(1,o,z)$, with a noise $z \sim p_0$, defines a deterministic mapping $a = \mu_\phi(o,z)$ from observation and noise to actions by ODEs. Since $z$ is stochastic, this induces a stochastic policy $\pi_\phi(a|o)$, enabling flow matching to serve as a generative policy model.

## 4 MULTI-AGENT COORDINATION VIA FLOW MATCHING (MAC-FLOW)

In this section, we introduce a novel MARL algorithm, dubbed MAC-Flow, which is a simple and expressive tool for extracting multi-agent policies via flow matching.

### 4.1 HOW DOES MAC-FLOW EXTRACT ONE-STEP POLICIES FOR COORDINATION?

**Our desiderata** are threefold: *(i)* to capture the distribution of coordinated behaviors, thereby preserving inter-agent dependencies under multi-agent dynamics; *(ii)* to ensure high practicality by

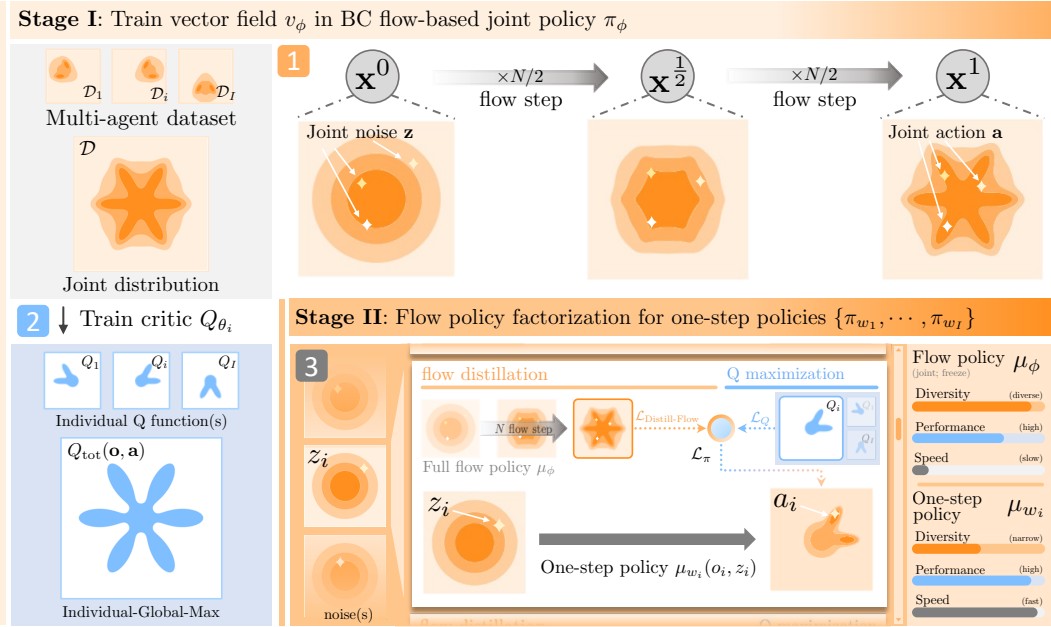

Figure 2: **Overview diagram of proposed solution.** Our solution, MAC-Flow, composes of two stages. The first stage models the joint action distribution via flow-matching to capture inter-agent dependencies, thereby facilitating the extraction of coordination behaviors more effectively than treating individual policies. For the next stage, individual critics are trained under the individual-global-max principle, thereby embedding behaviors for multi-agent coordination. At the second stage, practicality is highlighted by deriving individual policies for decentralized execution from a flow-based joint policy via Q maximization and BC distillation.

enabling decentralized execution and fast inference at test time; and *(iii)* to maintain algorithmic simplicity by avoiding unnecessary architectural overhead. Therefore, we adopt a two-stage strategy, which trains a joint policy via flow matching for *(i)*, then distills it into a set of individual policies for *(ii)*. Thanks to the simplicity of flow-matching and BC distillation, MAC-Flow directly fulfills *(iii)*.

**Overview.** Figure 2 shows an overview diagram for MAC-Flow. To achieve our goal, the first stage learns a joint observation- and time-dependent vector field $v_\phi(t, \mathbf{o}, \mathbf{z})$ to capture the multi-modal action distribution from the multi-agent dataset $\mathcal{D}$. This vector field serves as a joint policy $\mu_\phi(\mathbf{o}, \mathbf{z})$. Before proceeding to the next stage, we train individual critics $\{Q_{\theta_1}, \ldots, Q_{\theta_i}, \ldots, Q_{\theta_I}\}$ based on the IGM principle. In the second stage, we distill the flow-based joint policy into one-step sampling policies $\{\mu_{\phi_1}(o_1, z_1), \cdots, \mu_{w_i}(o_i, z_i) \cdots, \mu_{w_I}(o_I, z_I)\}$ for each agent $i$, where $w_i$ represents a parameter of an individual policy network for $i$-th agent. This relies on three key properties: **Definition** 4.1, the joint action distribution can be factorized into independent individual policies; **Proposition** 4.2, the mismatch between the joint distribution and its factorized approximation is upper-bounded by the distillation loss; and **Proposition** 4.3 the resulting performance gap is controlled via a Lipschitz bound on the value function. Together, our solution preserves the multi-modal structure of the joint policy while extracting individual policies for fast inference.

## 4.2 PROPOSED SOLUTION

**Joint Policy Extraction via Flow Matching.** The objective of the first stage is to build a flow-based joint policy $\mu_\phi(\mathbf{o}, \mathbf{z})$ via solely BC objective that accurately captures the joint action distribution in the offline multi-agent dataset $\mathcal{D}$. Concretely, we train it by expanding the flow-BC loss function $\mathcal{L}_{\text{Flow-BC}}$ (Equation 5) into the joint observation-action data sample as follows:

$$\mathcal{L}_{\text{Flow-BC}}(\phi) = \mathbb{E}_{\mathbf{x}^0 \sim \mathbf{p}_0, \, (\mathbf{o}, \mathbf{a}) \sim \mathcal{D}, \, t \sim \text{Unif}([0,1])} \left[ \| v_\phi(t, \mathbf{o}, \mathbf{x}^t) - (\mathbf{a} - \mathbf{x}^0) \|_2^2 \right] \quad (6)$$

where $\mathbf{x}^0 = [x_1^0, \cdots, x_I^0]$ and $\mathbf{p}_0 = \prod_{i=1}^{I} \mathcal{N}(0, \mathbf{I}_{d_i})$ denote the random sampled joint noise and the noise distributions for all agents. The trained vector field $v_\phi$ defines a joint flow $\psi_\phi(1, \mathbf{o}, \mathbf{z})$ and hence a stochastic joint policy $\pi_\phi(\mathbf{a} \mid \mathbf{o})$ through reparameterization with $\mathbf{z} \sim \mathbf{p}_0$.

**Flow-based Joint Policy Distillation.** Since execution under the CTDE framework must be fully decentralized, a joint policy conditioned on global observation is infeasible to deploy. We therefore factorize the flow-based joint policy into individual policies that approximate the individual action distribution while preserving coordination. This connection can be formalized by extending the IGM principle to the action distribution as follows.

**Definition 4.1** (Action distribution identical matching). *For a joint observation $\mathbf{o}$ and action $\mathbf{a}$ with agent-wise dimensions $d_i$, let $\pi(\mathbf{a} \mid \mathbf{o})$ be the joint action distribution. If each agent $i$ admits an individual distribution $\pi_i(a_i \mid o_i)$ such that $\pi(\mathbf{a} \mid \mathbf{o}) = \prod_{i=1}^{N} \pi_i(a_i \mid o_i)$, then we say that action distribution identical matching holds.*

This condition implies that decentralized execution via independent local sampling from $\pi_i(a_i \mid o_i)$ is distributionally equivalent to centralized execution of the joint policy $\pi(\mathbf{a} \mid \mathbf{o})$. In practice, to approximate this factorization, we introduce a *distillation loss* that aligns the product of individual policies with the flow-based joint policy as follows:

$$\mathcal{L}_{\text{Distill-Flow}}(\mathbf{w}) = \mathbb{E}_{\mathbf{o} \sim \mathcal{D}, \, \mathbf{z} \sim \mathbf{p}_0} \left[ \sum_{i=1}^{I} ||\mu_{w_i}(o_i, z_i) - [\mu_\phi(\mathbf{o}, \mathbf{z})]_i||_2^2 \right], \tag{7}$$

where $[\cdot]_i$ is the $i$-th subvector of joint variables, and $\mathbf{w}$ is the set of individual policy parameters $[w_1, \cdots, w_I]$. Importantly, distillation is not merely heuristic. Given both joint and factorized policies with the same noise, we obtain the following bound:

**Proposition 4.2** (2-Wasserstein upper bound of distillation). *Fix a joint observation $\mathbf{o}$. Let $\mathbf{z} \sim p_0$ be a noise variable, and define the joint policy mapping $\mu_\phi(\mathbf{o}, \mathbf{z}) \in \mathcal{A}$ and the factorized policy mapping $\mu_{\mathbf{w}}(\mathbf{o}, \mathbf{z}) = [\mu_{w_1}(o_1, z_1), \ldots, \mu_{w_I}(o_I, z_I)] \in \mathcal{A}$. Denote by $\pi_\phi(\mathbf{o})$ and $\pi_{\mathbf{w}}(\mathbf{o})$ the push-forward distributions of $\mathbf{p}_0$ through $\mu_\phi$ and $\mu_{\mathbf{w}}$, respectively. Then, the 2-Wasserstein distance between the joint policy and its factorization is upper-bounded by the square root of the distillation loss:*

$$W_2(\pi_{\mathbf{w}}(\mathbf{o}), \pi_\phi(\mathbf{o})) \leq \left( \mathbb{E}_{\mathbf{z} \sim p_0} \left[ ||\mu_{\mathbf{w}}(\mathbf{o}, \mathbf{z}) - \mu_\phi(\mathbf{o}, \mathbf{z})||_2^2 \right] \right)^{1/2}. \tag{8}$$

**Full Objective for Policy Factorization.** The goal of the second stage is to factorize a flow-based joint policy $\mu_\phi(\mathbf{o}, \mathbf{z})$ into a set of one-step sampling policies $\{\mu_{w_i}(o_1, z_1), \cdots, \mu_{w_I}(o_I, z_I)\}$ for $I$ agents under the IGM and Definition 4.1. Formally, the full loss function can be defined as follows:

$$\mathcal{L}_\pi(\mathbf{w}) = \mathbb{E}_{\mathbf{o} \sim \mathcal{D}, \, \mathbf{a} \sim \pi_{\mathbf{w}}, \, \mathbf{z} \sim \mathbf{p}_0} \left[ -Q_{\text{tot}}(\mathbf{o}, \mathbf{a}) + \alpha \sum_{i=1}^{I} ||(\mu_{w_i}(o_i, z_i) - [\mu_\phi(\mathbf{o}, \mathbf{z})]_i||_2^2 \right]. \tag{9}$$

As mentioned in Section 4.1, we design this function for one-step sampling policies to maximize the Q function and minimize the BC distillation losses. We train a critic network with a parameter $\theta_i$ for agent $i$. In practice, minimizing the $\mathcal{L}_\pi(\mathbf{w})$ in turn upper-bounds the performance difference in terms of the global value function $Q_{\text{tot}}$. The following proposition formalizes this guarantee.

**Proposition 4.3** (Lipschitz value gap bound). *Fix a joint observation $\mathbf{o}$ and assume $Q_{tot}(\mathbf{o}, \cdot)$ is $L_Q$-Lipschitz in the action: $\left| Q_{tot}(\mathbf{o}, \pi_\phi(\mathbf{o})) - Q_{tot}(\mathbf{o}, \pi_{\mathbf{w}}(\mathbf{o})) \right| \leq L_Q \|\mathbf{a}_\phi - \mathbf{a}_{\mathbf{w}}\|_2, \, \forall \mathbf{a}_\phi, \mathbf{a}_{\mathbf{w}} \in \mathcal{A}$. Denote by $\pi_\phi(\mathbf{o})$ and $\pi_{\mathbf{w}}(\mathbf{o})$ the push-forward distributions of the joint noise $\mathbf{p}_0$ through $\mu_\phi(\mathbf{o}, \cdot)$ and $\mu_{\mathbf{w}}(\mathbf{o}, \cdot) = [\mu_{\mathbf{w}_1}(o_1, z_1), \cdots, \mu_{\mathbf{w}_I}(o_I, z_I)]$, respectively. Then, the performance gap satisfies*

$$\left| \mathbb{E}_{\mathbf{a} \sim \pi_{\mathbf{w}}(\mathbf{o})} \left[ Q_{tot}(\mathbf{o}, \mathbf{a}) \right] - \mathbb{E}_{\mathbf{a} \sim \pi_\phi(\mathbf{o})} \left[ Q_{tot}(\mathbf{o}, \mathbf{a}) \right] \right| \leq L_Q \, W_2(\pi_{\mathbf{w}}(\mathbf{o}), \pi_\phi(\mathbf{o}))$$

$$\leq L_Q \sqrt{\left( \mathbb{E}_{\mathbf{z} \sim \mathbf{p}_0} \|\mu_{\mathbf{w}}(\mathbf{o}, \mathbf{z}) - \mu_\phi(\mathbf{o}, \mathbf{z})\|_2^2 \right)}. \tag{10}$$

For all mathematical derivations of the provided Propositions, please see Appendix D. Note that these properties collectively characterize bounded performance degradation under explicit assumptions rather than perfectly global optimality preservation.

### 4.3 DIDACTIC EXAMPLE: VALIDATING THE WASSERSTEIN–VALUE GAP RELATION

To further understand the theoretical idea, we study a toy example with grid world, a landmark covering task. This subsection aims to show how policy factorization, distributional mismatch, Wasserstein distances, and value gaps manifest during learning.

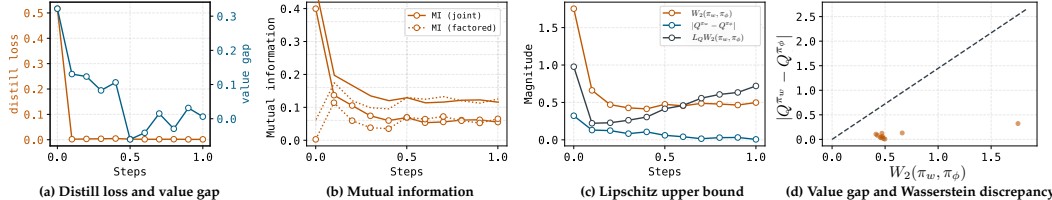

Figure 3: **Theoretical validation in a didactic example.** (*a*) Distillation loss and corresponding value gap between the joint and factored policies over training. (*b*) Inter-agent mutual information during training, comparing dependency strength in joint policy vs. factorized policies. (*c*) Empirical value gap alongside the theoretical Lipschitz upper bound predicted by Proposition 4.3. (*d*) Point-wise scatter of value gap versus Wasserstein discrepancy, showing all checkpoints lie below the theoretical bound.

**Didactic task setup.** In 2D plane environment, three agents aim to cover three fixed landmarks. Each agent observes its own position and all landmark positions, but not other agents. The action of agent $i$ is a movement vector $a^i \in \mathbb{R}^2$. The reward is defined through an optimal assignment. The reward is a negative distance to its assigned landmark. We provide visualization in Appendix H.4.

**Empirical evidence for propositions.** Proposition 4.2 and 4.3 relate the performance deviation between the joint policy and its factorized approximation to the distillation discrepancy measured in Wasserstein distance under controlled conditions. First, Figure 3 (*a*) plots both the distillation loss and the empirical value gap over training. As distillation proceeds, the value gap decreases in tandem with the loss, confirming the expected contraction. Second, improved distributional alignment results in reduced performance degradation. Next, Figure 3(*b*) shows that the joint policy exhibits strong inter-agent mutual information $\mathrm{MI}(a^i, a^j)$ during the early phase of training, reflecting the ambiguity in how agents initially partition the landmarks. As training progresses and the landmark assignments become more stable, this MI gradually decreases and eventually converges. In contrast, the factored policy starts from 0 MI, then gradually becomes similar to the joint policy. It maintains MI values below approximately 0.1 throughout training, which indicates that its independent parameterization cannot capture the interaction-induced dependencies without distillation. Finally, Figure 3 (*c*) shows that the empirical value gap stays below the theoretical upper bound at all checkpoints, with the gap tightening as learning stabilizes. A point-wise analysis in Figure 3 (*d*) further shows a clear monotonic trend: all samples lie beneath the linear envelope defined by $L_Q W_2$, indicating that the bound is both valid and informative in practice.

## 4.4 ALGORITHM SUMMARY

Algorithm 1 outlines `MAC-Flow`: we learn a BC flow-based joint policy $\mu_\phi(\mathbf{o}, \mathbf{z})$ via flow matching to model joint action distributions; train individual critics $\{Q_{\theta_i}\}$ under the IGM principle; and factorize $\mu_\phi$ into decentralized one-step policies $\{\mu_{w_i}\}$ via $Q$ guidance and BC distillation. During training, policy action for TD backups are sampled from $\mu_\phi(\mathbf{o}, \mathbf{z})$ using the Euler method (Algorithm 2), whereas at deployment, actions are generated directly by $\{\mu_{w_i}\}$. This design enhances inference speed while preserving coordinated expressivity, offering tractable value estimation under IGM, performance guarantees via Propositions 4.2 and 4.3, and single-step execution for fast inference.

## 5 EXPERIMENTS

The following subsection presents a suite of experiments designed to assess the effectiveness of `MAC-Flow` and experimental results via the following research questions (RQ) and answers.

**RQ1.** **How good is `MAC-Flow` over continuous and discrete action spaces of MARL benchmarks?**

**RQ2.** **How fast is the inference speed of `MAC-Flow` compared to diffusion-based solutions?**

**RQ3.** **Can `MAC-Flow` be extended beyond offline pretraining to online fine-tuning?**

**RQ4.** **How effective is the two-stage strategy in `MAC-Flow` framework?**

**RQ5.** **How effective is the IGM-based critic training in `MAC-Flow` framework?**

Please see Appendix F, G, and H to check additional RQs and a detailed description for experiments.

---

**Algorithm 1** `MAC-Flow`

**while** not converged **do**

  Sample batch $\{(o_i, a_i, r_i, o'_i)\}_{i=1}^I \sim \mathcal{D}$

  # TRAIN BC FLOW-BASED JOINT POLICY $\pi_\phi$
  Set variables $\mathbf{x}^0 \leftarrow \mathbf{z} \sim \mathbf{p}^0, \mathbf{x}^1 \leftarrow \mathbf{a}, t \sim \mathrm{Unif}([0,1])$
  Calculate noise point $\mathbf{x}_t \leftarrow (1-t)\mathbf{x}_0 + t\mathbf{x}_1$
  Update $\phi$ using Equation (6)

  # TRAIN INDIVIDUAL CRITIC $Q_{\theta_i}$
  **for** $i = 1, \cdots, I$ **do**
    Sample noise $x_i^0 \leftarrow z_i \sim p_i^0$
  Sample joint action $\mathbf{a}' \leftarrow \mu_\phi(\mathbf{o}', \mathbf{x}^0)$     ▷ Algorithm 2
  Update $\{\theta_i\}_{i=1}^I$ by $\mathbb{E}[Q_{\theta_i}(o_i, a_i) - r_i - \gamma Q_{\bar{\theta}_i}(o'_i, a'_i)]$

  # EXTRACT INDIVIDUAL POLICY $\pi_{w_i}$
  **for** $i = 1, \cdots, I$ **do**
    Sample noise $x_i^0 \leftarrow z_i \sim p_i^0$
    Sample action $a_i \leftarrow \mu_{w_i}(o_i, x_i^0)$
  Update $\{w_i\}_{i=1}^I$ using Equation (9)

  **return** set of one-step policies $\{\pi_{w_1}, \cdots, \pi_{w_I}\}$

---

**Algorithm 2** Sampling

**function** $\mu_\phi(\mathbf{o}, \mathbf{x})$
$d \leftarrow 1/M$
$t \leftarrow 0$
**for** $k \in \{0, \cdots, M-1\}$ **do**
  $\mathbf{x} \leftarrow \mathbf{x} + v_\phi(t, \mathbf{o}, \mathbf{x})d$
  $t \leftarrow t + d$
**return** $\mathbf{x}$

---

## 5.1 ENVIRONMENTAL SETUPS

We evaluate the proposed solution, `MAC-Flow`, on four widely used MARL environments: StarCraft multi-agent challenge (SMAC) v1, SMACv2, multi-agent MuJoCo (MA-MuJoCo), and the multiple-particle environment (MPE). A description of the testbeds and datasets is provided below.

**SMACv1** (*discrete action*) provides a real-time combat environment where two teams compete, with one controlled by built-in AI and the other by learned policies. It incorporates both homogeneous and heterogeneous unit settings, thereby enabling diverse strategic coordination requirements. For **offline datasets**, we use the assets from off-the-grid benchmark (Formanek et al., 2023), *including* three quality datasets for each map, *e.g.*, Good, Medium, and Poor.

**SMACv2** (*discrete action*) extends SMACv1 by addressing the limited randomness of SMACv1 through three major modifications: randomized start positions, randomized unit types, and adjusted unit sight and attack ranges. These changes increase the diversity of the scenarios, making the tasks more challenging. For **offline datasets**, we use the assets from off-the-grid benchmark (Formanek et al., 2023), *including* a dataset for each map, *e.g.*, Replay.

**MA-MuJoCo** (*continuous action*) decomposes single robotic systems into multiple agents, each responsible for controlling a specific subset of joints. This design enables agents to coordinate in achieving shared objectives. For **its datasets**, we leverage the asset from Wang et al. (2023), *including* four datasets for each robotic control, *e.g.*, Expert, Medium-Expert, Medium, and Medium-Replay.

**MPE** (*continuous action*) is a lightweight benchmark commonly used for studying cooperative coordination. Agents are represented as particles moving in a two-dimensional continuous space, where they must coordinate to achieve goals. We leverage the offline datasets collected by Pan et al. (2022), *including* four quality datasets, *e.g.*, Expert, Medium, Medium-Replay, and Random.

**Baselines**. For offline MARL experiments, we use the three categories for baselines. For Gaussian policies, we consider the extension of SARL, *e.g.*, BC, BCQ (Fujimoto et al., 2019), CQL (Kumar et al., 2020), and TD3BC (Fujimoto & Gu, 2021), and standard offline MARL solutions, *e.g.*, ICQ (Yang et al., 2021), OMAR (Pan et al., 2022), and OMIGA (Wang et al., 2023). For diffusion policies, we select recent offline MARL algorithms, *e.g.*, diffusion BC, MADiff (Zhu et al., 2024), and DoF (Li et al., 2025a). Lastly, the flow policies include Flow BC and our proposed solution.

## 5.2 EXPERIMENTAL RESULTS AND RESEARCH Q&A

**Contribution overview with RQ1 and RQ2**. Diffusion baselines, *e.g.*, DoF, demonstrate strong coordination performance, but they incur substantial cost due to iterative denoising process. In contrast, `Mac-Flow` trades a small amount of expressiveness for dramatically faster optimization.

Table 1: **Performance evaluation for discrete action control**. We present a performance comparison across 2 benchmarks, 8 tasks, and 18 datasets. These results are averaged over 6 seeds, and we report the two standard deviations after the $\pm$ sign. We highlight the best performance in **bold** and the second best in underlined.

| | Scenarios | Dataset | Gaussian policies | | | Diffusion policies | | | Flow policies | |
|---|---|---|---|---|---|---|---|---|---|---|
| | | | BC | MABCQ | MACQL | Diffusion BC | MADiff | DoF | Flow BC | MAC-Flow |
| SMACv1 | 3m | Good | 16.0 ±1.0 | 3.7 ±1.1 | 19.1 ±0.1 | 19.5 ±0.5 | 19.3 ±0.5 | 19.8 ±0.2 | **20.0 ±0.0** | 19.8 ±0.2 |
| | | Medium | 8.2 ±0.8 | 4.0 ±1.0 | 13.7 ±0.3 | 13.3 ±0.7 | 16.4 ±2.6 | **18.6 ±1.2** | 14.7 ±1.5 | 18.0 ±3.2 |
| | | Poor | 4.4 ±0.1 | 3.4 ±1.0 | 4.2 ±0.1 | 4.2 ±0.2 | 10.3 ±6.1 | **10.9 ±1.1** | 4.5 ±0.1 | 10.6 ±2.2 |
| | 8m | Good | 16.7 ±0.4 | 4.8 ±0.6 | 18.9 ±0.9 | 19.4 ±0.5 | 18.9 ±1.1 | 19.6 ±0.3 | 19.5 ±0.2 | **19.7 ±0.3** |
| | | Medium | 10.7 ±0.5 | 5.6 ±0.6 | 15.5 ±1.5 | 18.6 ±0.6 | 16.8 ±1.6 | 18.6 ±0.8 | 18.2 ±0.8 | **19.4 ±0.6** |
| | | Poor | 5.3 ±0.1 | 3.6 ±0.8 | 7.5 ±1.0 | 4.8 ±0.2 | 9.8 ±0.9 | **12.0 ±1.2** | 4.9 ±0.1 | 11.5 ±0.8 |
| | 2s3z | Good | 18.2 ±0.4 | 7.7 ±0.9 | 17.4 ±0.3 | 18.0 ±1.0 | 15.9 ±1.2 | 18.5 ±0.8 | **19.5 ±0.1** | 19.5 ±0.5 |
| | | Medium | 12.3 ±0.7 | 7.6 ±0.7 | 15.6 ±0.4 | 13.4 ±1.4 | 15.6 ±0.3 | **18.1 ±0.9** | 15.1 ±2.0 | 17.6 ±0.6 |
| | | Poor | 6.7 ±0.3 | 6.6 ±0.2 | 8.4 ±0.8 | 6.2 ±1.2 | 8.5 ±1.3 | **10.0 ±1.1** | 6.9 ±0.8 | 8.5 ±0.6 |
| | 5m_vs_6m | Good | 15.8 ±3.6 | 2.4 ±0.4 | 16.2 ±1.6 | 16.8 ±2.3 | 16.5 ±2.8 | 17.7 ±1.1 | 14.7 ±2.1 | **18.6 ±3.5** |
| | | Medium | 12.4 ±0.9 | 3.8 ±0.5 | 15.1 ±2.9 | 12.5 ±2.1 | 15.2 ±2.6 | **16.2 ±0.9** | 12.8 ±0.8 | 15.6 ±1.3 |
| | | Poor | 7.5 ±0.2 | 3.3 ±0.5 | 10.5 ±3.1 | 8.0 ±1.0 | 8.9 ±1.3 | **10.8 ±1.0** | 7.7 ±0.8 | 9.8 ±2.1 |
| | 2c_vs_64zg | Good | 17.5 ±0.4 | 10.1 ±0.2 | 12.9 ±0.2 | 17.8 ±1.3 | 14.7 ±2.2 | 16.1 ±0.8 | 18.0 ±1.3 | **19.1 ±0.8** |
| | | Medium | 12.5 ±0.3 | 9.9 ±0.2 | 11.6 ±0.1 | 10.5 ±1.1 | 12.8 ±1.2 | 13.9 ±0.9 | 11.8 ±2.6 | **14.9 ±4.1** |
| | | Poor | 9.7 ±0.2 | 9.0 ±0.2 | 10.2 ±0.1 | 10.2 ±2.3 | 10.8 ±1.1 | **11.5 ±1.1** | 10.0 ±0.3 | 11.4 ±0.4 |
| | **Average rewards** | | 12.2 | 5.5 | 13.1 | 13.0 | 13.8 | **15.6** | 13.4 | **15.6** |
| SMACv2 | terran_5_vs_5 | Replay | 7.3 ±1.0 | 13.8 ±4.4 | 11.8 ±0.9 | 9.3 ±0.9 | 13.3 ±1.8 | 15.4 ±1.3 | 8.3 ±1.9 | **16.6 ±4.3** |
| | zerg_5_vs_5 | Replay | 6.8 ±0.6 | 10.3 ±1.2 | 10.3 ±3.4 | 8.1 ±1.7 | 10.2 ±1.1 | **12.0 ±1.1** | 4.6 ±0.5 | 9.8 ±1.5 |
| | terran_10_vs_10 | Replay | 7.4 ±0.5 | 12.7 ±2.0 | 11.8 ±2.0 | 5.5 ±1.5 | 13.8 ±1.3 | **14.6 ±1.1** | 5.8 ±1.7 | 13.0 ±4.7 |
| | **Average rewards** | | 7.2 | 12.3 | 11.3 | 7.6 | 12.4 | **14.0** | 6.2 | 13.1 |

Table 2: **Performance evaluation for continuous action control**. We present a performance comparison across 2 benchmarks, 4 tasks, and 16 datasets. Results are reported following the conventions of Table 1. For readability, we use the acronyms *M-E* and *M-R* for Medium-Expert and Medium-Replay, respectively.

| | Scenarios | Dataset | Extension of offline SARL | | Offline MARL | | | | |
|---|---|---|---|---|---|---|---|---|---|
| | | | MATD3BC | MACQL | ICQ | OMAR | OMIGA | MADiff | MAC-Flow |
| MA-MuJoCo | HalfCheetah | Expert | 4401.6 ±169.1 | 4589.5 ±98.5 | 2955.9 ±459.2 | −206.7 ±161.1 | 3383.6 ±552.7 | **4711.4 ±213.6** | 4650.0 ±271.6 |
| | | Medium | 2620.8 ±69.9 | 3189.4 ±306.9 | 2549.3 ±96.3 | −265.7 ±147.0 | 3608.1 ±237.4 | 2650.0 ±365.4 | **4358.5 ±369.2** |
| | | M-R | **3528.9 ±120.9** | 3500.7 ±293.9 | 1922.4 ±612.9 | −235.4 ±154.9 | 2504.7 ±83.5 | 2830.5 ±292.8 | 3030.2 ±436.8 |
| | | M-E | 3518.1 ±381.0 | 4738.2 ±181.1 | 2834.0 ±420.3 | −253.8 ±63.9 | 2948.5 ±518.9 | 4410.9 ±836.8 | **5139.9 ±84.1** |
| | Hopper | Expert | 3309.9 ±4.5 | 3359.1 ±513.8 | 754.7 ±806.3 | 2.4 ±1.5 | 859.6 ±709.5 | 2853.3 ±593.8 | **3592.1 ±8.9** |
| | | Medium | 870.4 ±156.7 | 901.3 ±199.9 | 501.8 ±14.0 | 21.3 ±24.9 | 1189.3 ±544.3 | **1436.8 ±449.5** | 1023.5 ±253.0 |
| | | M-R | 269.7 ±41.8 | 31.4 ±15.2 | 195.4 ±103.6 | 3.3 ±3.2 | 774.2 ±494.3 | 936.1 ±574.0 | **1166.3 ±451.9** |
| | | M-E | 2904.3 ±477.4 | 2751.8 ±123.3 | 355.4 ±373.9 | 1.4 ±0.9 | 709.0 ±595.7 | 2810.4 ±723.2 | **2988.3 ±480.2** |
| | Ant | Expert | 2046.9 ±17.1 | **2082.4 ±21.7** | 2050.0 ±11.9 | 312.5 ±297.5 | 2055.5 ±1.6 | 2060.0 ±10.3 | 2060.2 ±20.0 |
| | | Medium | 1422.6 ±21.1 | 1033.9 ±66.4 | 1412.4 ±10.9 | −1710.0 ±1589.0 | 1418.4 ±5.4 | 1428.4 ±14.7 | **1432.4 ±17.8** |
| | | M-R | 995.2 ±52.8 | 434.6 ±108.3 | 1016.7 ±53.5 | −2014.2 ±844.7 | 1105.1 ±88.9 | 1294.5 ±360.2 | **1498.4 ±20.3** |
| | | M-E | 1636.1 ±96.0 | 1800.2 ±21.5 | 1590.2 ±85.6 | −2992.8 ±7.0 | 1720.3 ±110.6 | 1740.2 ±158.9 | **2053.3 ±20.4** |
| | **Average rewards** | | 2293.7 | 2367.7 | 1511.5 | −611.5 | 1856.4 | 2430.21 | **2749.4** |
| MPE | Spread | Expert | 108.3 ±3.3 | 98.2 ±5.2 | **114.9 ±2.6** | 104.0 ±3.4 | 80.8 ±13.8 | 95.0 ±5.3 | 101.7 ±10.9 |
| | | Medium | 29.3 ±4.8 | 34.1 ±7.2 | 47.9 ±18.9 | 29.3 ±5.5 | 30.1 ±16.9 | 64.9 ±7.7 | **80.1 ±20.6** |
| | | M-R | 15.4 ±5.6 | 20.0 ±8.4 | 37.9 ±12.3 | 13.6 ±5.7 | 5.4 ±11.0 | 30.3 ±2.5 | **50.4 ±33.2** |
| | | Random | 9.8 ±4.9 | 24.0 ±9.8 | **34.4 ±5.3** | 6.3 ±3.5 | −3.8 ±12.3 | 6.9 ±3.1 | 31.1 ±6.8 |
| | **Average rewards** | | 40.7 | 44.1 | 58.8 | 38.3 | 28.1 | 49.2 | **65.8** |

Additionally, `MAC-Flow` incurs a modest increase in training time over Gaussian models, but still trains far faster than diffusion baselines. Full results are provided in Appendix H.2

**A1**: For **RQ1** (*Performance*), `MAC-Flow` achieves best or second-best average performance across four benchmarks regardless of continuous or discrete action space. Table 1 and 2 summarize the performance comparison across four benchmarks. These results demonstrate that `MAC-Flow` matches the performance of `DoF` by combining expressive flow-based modeling of joint action distribution with an efficient distillation step into decentralized one-step policies, thereby preserving coordination quality while ensuring scalability across discrete and continuous benchmarks.

**A2**: For **RQ2** (*Inference speed*), `MAC-Flow` achieves averaged ×14.5 faster inference than diffusion policies with comparable performance. Figure 4 shows its faster inference while maintaining competitive performance relative to `MADiff` and `DoF`. Moreover, `MAC-Flow` matches the inference speed of prior offline MARL algorithms but significantly outperforming them in performance. Theoretically, the per-agent inference complexity of `MAC-Flow` is low, especially $\mathcal{O}(1)$, in contrast to $\mathcal{O}(K)$ for `DoF` and $\mathcal{O}(IK)$ for `MADIFF`, where $K$ is the diffusion steps and $I$ is the number of agents. Full details are in Appendix C.

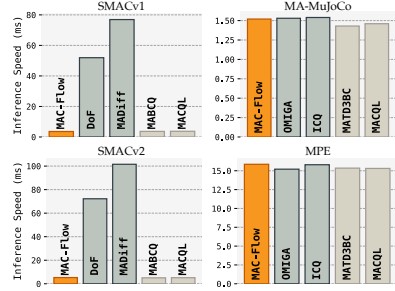

Figure 4: **Inference time.** These results are averaged over each benchmark's scenarios.

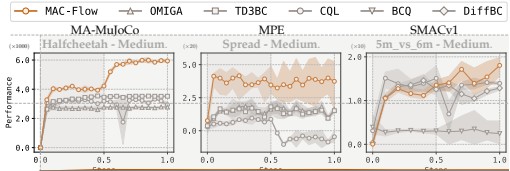 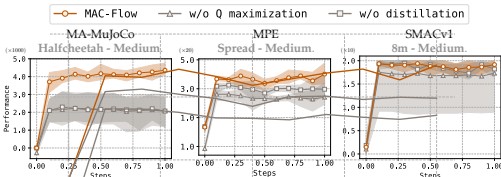

Figure 5: **Offline-to-online experiments.** Online fine-tuning starts at 0.5 normalized steps.

Figure 6: **Ablation study for RQ4.** We test the effect of the distillation phase and its RL objective.

**A3**: For **RQ3** (*Offline-to-online*) , `MAC-Flow` can seamlessly use online rollouts to fine-tune itself, enabling it to achieve better results than previous methods. Figure 5 shows the training curves of `MAC-Flow` and previous baselines, where online fine-tuning begins after 0.5 iteration steps. Some baselines fail to account for exploration and thus remain limited to their offline performance, while `MA-CQL` exhibits the reported issue of a sharp performance drop at the initial stage of the online phase. In contrast, our approach effectively improves performance in the online stage by leveraging newly collected rollouts. In practice, `MAC-Flow` can fine-tune its networks under the same objective used in offline learning by simply augmenting the offline dataset with additional online rollouts.

**A4**: For **RQ4** (*Ablation on distillation with Q maximization*) , both the full objective and training scheme of `MAC-Flow` are essential. Figure 6 presents the ablation study on Q maximization in Equation (9) and the distillation stage in the two-stage training scheme. Specifically, `w/o Q maximization` corresponds to a one-step sampling policy without an RL objective, while `w/o distillation` refers to a pure BC flow policy that requires an ODE solver. Across all scenarios, `MAC-Flow` consistently outperforms its ablated counterparts, demonstrating that removing either Q-maximization or the distillation phase substantially limits policy learning. These results highlight that such components are critical to achieving strong performance across diverse tasks.

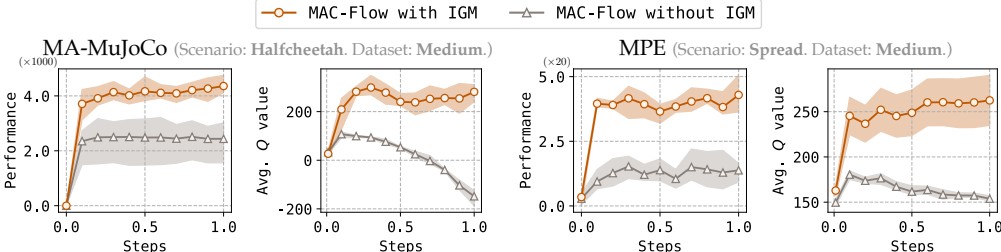

Figure 7: **Ablation study for RQ5.** This figure shows the learning curves for performance and $Q$ value. Performance's y-axes of MA-MuJoCo and MPE are scaled as 1000 and 20 units, respectively.

**A5**: For **RQ5** (*Ablation on IGM*) , IGM-based critic training is crucial for ensuring stability in critic optimization, consistency in $Q$-value estimation, and superior performance in the `MAC-Flow` framework. Figure 7 presents learning curves for the ablation study comparing `MAC-Flow` with IGM and without IGM. The IGM-based variant achieves substantially higher performance, while the non-IGM counterpart stagnates at suboptimal levels. In particular, IGM leads to significantly lower and more stable loss curves in a multi-agent setting; additionally, the non-IGM baseline exhibits a collapse of $Q$ estimates as training progresses.

## 6 CONCLUSIONS

In this work, we propose `MAC-Flow`, a novel MARL algorithm that learns a flow-based joint policy to capture the multi-modality of multi-agent datasets, and then distills it into decentralized one-step sampling policies using a combination of RL and BC objectives. Our experiments show that `MAC-Flow` addresses the trade-off between inference efficiency and performance in offline MARL.

**Future Directions and Impact Statement.** While our approach demonstrates strong performance and efficiency gains, extending `MAC-Flow` to more diverse and dynamic environments remains an important direction. We should develop an algorithm that can integrate other pre-trained distributions to enhance the diversity of decentralized policies. This can enable more flexible role adaptation and generalizability improvement, ensuring that agents adapt to new scenarios. Such directions increase the stability of multi-agent systems. This line of research provides a foundation for future advances in generalizable MARL and its deployment in real-world domains.

ACKNOWLEDGEMENTS

This work was supported by the National Science Foundation Graduate Research Fellowship Program under Grant No. NSF 2340651, NSF 2402650, DARPA HR00112490431, and ARO W911NF-24-1-0193. This research used the Texas Advanced Computing Center (TACC) at The University of Texas at Austin for providing computational resources.

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

# Appendix

## CONTENTS

# A MISCELLANEOUS

## A.1 SUMMARY OF NOTATIONS

Dec-POMDP elements

| Notation | Description | Notation | Description |
|---|---|---|---|
| $\mathcal{I}$ | set of agents | $i$ | agent index |
| $I$ | number of agents | $\gamma \in [0,1)$ | discount factor |
| $\mathcal{S}$ | global state space | $s^t$ | state at time $t$ |
| $\mathcal{O}_i$ | observation space of agent $i$ | $o_i^t$ | local observation of agent $i$ |
| $\mathcal{A}_i$ | action space of agent $i$ | $a_i^t$ | action of agent $i$ |
| $\mathcal{T}$ | state transition function | $\Omega_i$ | observation function of agent $i$ |
| $r_i$ | reward function of agent $i$ | $R_i^t$ | return of agent $i$ at time $t$ |
| $\tau$ | trajectory | $\mathcal{D}$ | offline dataset (replay buffer) |

Algorithm elements (Flow Matching / Policies)

| Notation | Description | Notation | Description |
|---|---|---|---|
| $p_0$ | prior noise distribution | $p_1$ | target action distribution |
| $x_0$ | noise sample from $p_0$ | $x_1$ | target action sample |
| $x_t$ | interpolated point $(1-t)x_0 + tx_1$ | $t$ | continuous flow step |
| $v_\phi(t,o,x)$ | velocity field conditioned on $o$ | $\phi(t,x)$ | flow trajectory |
| $\mathbf{z}$ | joint noise variable | $z_i$ | noise for agent $i$ |
| $\mu_\phi(\mathbf{o},\mathbf{z})$ | flow-based joint policy | $\pi_\phi(\mathbf{a}|\mathbf{o})$ | induced stochastic joint policy |
| $\mu_{w_i}(o_i,z_i)$ | one-step policy for agent $i$ | $\pi_{w_i}(a_i|o_i)$ | induced one-step sampling policy |
| $Q_{\text{tot}}(o,a)$ | global Q-function | $Q_{\theta_i}(o_i,a_i)$ | individual Q-function |

RL Training

| Notation | Description | Notation | Description |
|---|---|---|---|
| $\theta$ | critic parameters | $\bar{\theta}$ | target critic parameters |
| $\phi$ | BC flow policy parameters | $w = \{w_i\}$ | parameters of one-step policies |
| $\eta$ | learning rate | $B$ | batch size |
| $K$ | number of updates | $T$ | episode horizon |
| $\alpha$ | balancing coefficient (regularization) | $f(\cdot,\cdot)$ | divergence measure |

## B EXTENSIVE RELATED WORKS

**Behavioral-regularized Actor Critic.** A closely related line of our research is behavioral-regularized actor-critic (BRAC) (Wu et al., 2019). The BRAC is one of the simplest and most powerful policy extraction strategies among offline RL solutions (Park et al., 2024). These approaches solve the out-of-distribution sample issue by constraining the learned policy to remain close to the behavior policy. In general, this is implemented by adding divergence penalties or regularization terms in the actor and critic updates, thereby stabilizing policy improvement in offline settings (Tarasov et al., 2023; Fujimoto & Gu, 2021; Kostrikov et al., 2021). Although these methods are primarily designed for single-agent RL (SARL) and rely on relatively simple Gaussian policies to model the action space, they are often adopted as baselines in MARL by being naively extended to multi-agent settings (Pan et al., 2022; Formanek et al., 2023; Wang et al., 2023; Zhu et al., 2024; Li et al., 2025a; Qiao et al., 2025). As a result, they struggle to capture the multi-modality of joint action distributions that are inherent in cooperative MARL scenarios, thereby being behind on their performance compared to advanced architecture-based RL solutions.

In contrast, our proposed solution embraces the same principle of behavioral regularization, striking a balance between $Q$ maximization and fidelity to the offline dataset. `MAC-Flow`, specifically, leverages flow-matching to extract a rich generative model of the joint action space, and then introduces a distillation loss analogous to the behavioral penalty in BRAC at the individual policy extraction phase. This allows decentralized one-step policies to inherit both expressiveness from the flow-based policy and regularization from the dataset distribution, concurrently guaranteeing coordination in multi-agent settings.

**Short-cut Diffusion or Flow.** Another relevant line and motivation of this work is shortcut generative modeling, *e.g.*, shortcut diffusion and shortcut flow matching (Frans et al., 2024; Park et al., 2025). These approaches alleviate the inefficiency of iterative generative models (denoising diffusion or ODE/SDE solvers) (Sohl-Dickstein et al., 2015; Ho et al., 2020; Song & Ermon, 2019; Song et al., 2020b; Nichol & Dhariwal, 2021) by either reducing the number of denoising steps (Song et al., 2020a; Salimans & Ho, 2022; Meng et al., 2023a; Song et al., 2023) or by learning direct mappings that approximate the multi-step generative process with fewer evaluations (Frans et al., 2024; Luhman & Luhman, 2021; Lipman et al., 2022). In the context of RL, shortcut generative methods have been explored to accelerate policy sampling while retaining the expressive capacity of diffusion or flow models to address complicated problems.

Our work shares the motivation of achieving fast inference with expressive policies, but introduces a key adaptation for the multi-agent setting. Rather than merely shortening generative trajectories, `MAC-Flow` factorizes the flow-based joint policy into decentralized one-step sampling policies, supported by theoretical guarantees under the IGM principle. This perspective extends shortcut flow approaches, where the shortcut lies not only in time complexity but also in the structural factorization of multi-agent policies. By combining flow-based modeling with policy distillation guided by the IGM principle, `MAC-Flow` generalizes shortcut generative techniques to address the scalability and coordination challenges unique to offline MARL.

## C  COMPLEXITY ANALYSIS

To theoretically support the empirical inference efficiency of `MAC-Flow`, we provide a big-$\mathcal{O}$ analysis comparing its per-agent and total inference-time complexity against two diffusion-based SOTA baselines, `DoF` and `MADiff`. We analyze the computational complexity of generating one joint action at a single environment step.

**Setups.** For this discussion, let $I$ denote the number of agents and $T$ the number of iterative steps required by diffusion or flow policies. The input dimensions for per-agent observation and action are treated as $d_{o_i}, d_{a_i} = \mathcal{O}(1)$. Although our solution is based on a simple $[512, 512, 512, 512]$-sized MLP network, and `MADiff` and `DoF` employ a U-net-based temporal architecture, we posit that all constant factors are absorbed in asymptotic notation.

Table 3: **Asymptotic inference time complexity analysis.** This table reports big-$\mathcal{O}$ analysis about input dimension, per-agent cost, and total cost of producing one joint action at a single environment step.

| Method | Decentralizeation | Input dimension | Per-agent complexity | Total complexity |
|---|---|---|---|---|
| MAC-Flow | Yes | $\mathcal{O}(1)$ | $\mathcal{O}(1)$ | $\mathcal{O}(I)$ |
| DoF | Yes | $\mathcal{O}(1)$ | $\mathcal{O}(K)$ | $\mathcal{O}(IK)$ |
| MADiff-D | Yes | $\mathcal{O}(I)$ | $\mathcal{O}(IK)$ | $\mathcal{O}(I^2K)$ |
| MADiff-C | No | $\mathcal{O}(I)$ | $\mathcal{O}(IK)$ | $\mathcal{O}(IK)$ |

**MAC-Flow.** Our solution extracts decentralized one-step sampling policies for each agent. Therefore, at the inference phase, each agent can produce its action with a single forward pass using simple MLP networks. The per-agent complexitiy is $\mathcal{O}(1)$, and the total complexity is $\mathcal{O}(I)$.

**DoF.** This decomposes the centralized diffusion process into decentralized per-agent processes. Each agent must execute a full $K$-step denoising chain, although the factorization ensures that the per-step input dimension is $\mathcal{O}(1)$. independent of $I$. Thus, the per-agent complexity is $O(K)$ and the total complexity is $O(IK)$. We follow the default setups, `DoF`-Trajectory, $W$-Concat factorization, and 200-steps denoising iterations.

**MADiff.** We report two variations for `MADiff`, centralized and decentralized versions. For `MADiff-D`, each agent conditions the diffusion model on its own local observation. Nevertheless, due to the model's architecture, the diffusion process generates a full joint trajectory that includes both the agent itself and its teammates. As a result, the input size at each denoising step scales with the number of agents, *i.e.*, $\mathcal{O}(I)$. Moreover, since the denoising process requires $K$ iterative steps, the per-agent complexity is $\mathcal{O}(IK)$. As all $I$ agents must perform this computation independently, the total complexity reaches $\mathcal{O}(I^2K)$, which highlights a quadratic dependence on the number of agents. Next, for `MADiff-C`, a single diffusion model jointly generates the actions for all agents in one forward chain. At each denoising step, the input dimension remains $\mathcal{O}(I)$, and the process requires $K$ iterations, yielding a per-agent complexity of $\mathcal{O}(IK)$. However, because the joint model executes once for all agents, the total complexity is limited to $\mathcal{O}(IK)$. This distinction emphasizes that `MADiff-C` avoids the quadratic blow-up observed in `MADiff-D`, albeit at the cost of requiring centralized execution and communication during deployment.

**Discussion Summary.** Table 3 summarizes the theoretical inference complexities. Our analysis shows that `MAC-Flow` achieves constant-time inference per agent, independent of both $I$ and $K$, thereby scaling linearly with the number of agents. In contrast, diffusion-based methods inherit the iterative overhead of $K$ denoising steps. While `DoF` alleviates input scaling via factorization, its complexity remains $\mathcal{O}(IK)$. `MADiff-C` also maintains $\mathcal{O}(IK)$ complexity but requires centralized execution. Finally, `MADiff-D` is the least efficient, with a quadratic dependence on the number of agents due to per-agent joint inference, resulting in $\mathcal{O}(I^2K)$. These asymptotic distinctions theoretically underpin the empirical findings reported in the main text, where `MAC-Flow` demonstrates a $13.7 \sim 21.4\times$ speedup over diffusion-based baselines.

# D  MATHEMATICAL DERIVATION

This section provides a mathematical derivation for two propositions. Before looking deeper into them, we first show the Lemma related to the comparability of joint and factorized policies.

**Lemma D.1** (Comparability of Joint and Factorized Policies). *Let $\mathbf{o}$ be a joint observation and let $\mathbf{z} \sim \mathbf{p}_0$ denote the joint noise variable. Consider the flow-based joint mapping $\mu_\phi(\mathbf{o}, \mathbf{z})$ that induces the push-forward distribution $\pi_\phi(\mathbf{o})$, and the factorized mapping $\mu_w(\mathbf{o}, \mathbf{z}) = [\mu_{w_1}(o_1, z_1), \ldots, \mu_{w_I}(o_I, z_I)]$ that induces $\pi_w(o)$. By Definition 4.1, if action distribution identical matching holds, then the joint distribution can be factorized as a product of individual policies. Even when exact matching does not hold, both $\pi_\phi(\mathbf{o})$ and $\pi_w(\mathbf{o})$ are defined as push-forward distributions of the same base noise $\mathbf{p}_0$. Hence, $\pi_\phi(\mathbf{o}, )$ and $\pi_w(\mathbf{o})$ are comparable within the same probability space, and their discrepancy can be measured via a divergence function.*

## D.1  PROOF FOR PROPOSITION 4.2

Let $\mathbf{o}$ be a joint observation and $\mathbf{z} \sim \mathbf{p}_0$ a joint noise variable. Define the joint policy mapping $\mu_\phi(\mathbf{o}, \mathbf{z}) \in \mathcal{A}_1 \times \cdots \times \mathcal{A}_\mathcal{I}$ and the factorized policiy mapping $\mu_w(\mathbf{o}, \mathbf{z}) = [\mu_{w_1}(o_1, z_1), \cdots, \mu_{w_I}(o_I, z_I)]$. Denote by $\mathbf{a} \sim \pi_\phi(\mathbf{o})$ and $\mathbf{a} \sim \pi_w(\mathbf{o})$ the push-forward distributions of $\mathbf{p}_0$ through $\mu_\phi$ and $\mu_w$, respectively.

Following Lemma D.1, the squared 2-Wasserstein distance is defined as

$$W_2^2\left(\pi_w(\mathbf{o}), \pi_\phi(\mathbf{o})\right) = \inf_{\lambda \in \Lambda(\pi_w, \pi_\phi)} \mathbb{E}_{(\mathbf{a}, \mathbf{y}) \sim \lambda}\left[\|\mathbf{a} - \mathbf{y}\|_2^2\right],$$

where $\Lambda(\pi_w, \pi_\phi)$ denotes the set of coupling distributions between $\pi_w$ and $\pi_\phi$.

By choosing the specific coupling $\lambda$ induced by sampling $\mathbf{z} \sim \mathbf{p}_0$ and pairing $\mu_w(\mathbf{o}, \mathbf{z})$ with $\mu_\phi(\mathbf{o}, \mathbf{z})$, we obtain

$$W_2^2\left(\pi_w(\mathbf{o}), \pi_\phi(\mathbf{o})\right) \leq \mathbb{E}_{\mathbf{z} \sim \mathbf{p}_0}\left[\|\mu_w(\mathbf{o}, \mathbf{z}) - \mu_\phi(\mathbf{o}, \mathbf{z})\|_2^2\right]$$

.

Then, taking square roots on both sides yields the desired inequality:

$$W_2\left(\pi_w(\mathbf{o}), \pi_\phi(\mathbf{o})\right) \leq \left(\mathbb{E}_{\mathbf{z} \sim \mathbf{p}_0}\left[\|\mu_w(\mathbf{o}, \mathbf{z}) - \mu_\phi(\mathbf{o}, \mathbf{z})\|_2^2\right]\right)^{1/2}.$$

The proof is completed.  □

## D.2  PROOF FOR PROPOSITION 4.3

Fix a joint observation $\mathbf{o}$ and assume the global Q-function $Q_{\text{tot}}(\mathbf{o}, \cdot)$ is $L_Q$-Lipschitz in its action argument, *i.e.*,

$$|Q_{\text{tot}}(\mathbf{o}, \mathbf{a}) - Q_{\text{tot}}(\mathbf{o}, \mathbf{y})| \leq L_Q \|\mathbf{a} - \mathbf{y}\|_2, \quad \forall \mathbf{a}, \mathbf{y} \in \mathcal{A}_1 \times \cdots \times \mathcal{A}_\mathcal{I}.$$

Let $\pi_\phi(\mathbf{o})$ and $\pi_w(\mathbf{o})$ denote the push-forward distributions of $p_0$ under $\mu_\phi(\mathbf{o}, \cdot)$ and $\mu_w(\mathbf{o}, \cdot)$, respectively. Then,

$$\left|\mathbb{E}_{\mathbf{a} \sim \pi_w(\mathbf{o})}[Q_{\text{tot}}(\mathbf{o}, \mathbf{a})] - \mathbb{E}_{\mathbf{a} \sim \pi_\phi(\mathbf{o})}[Q_{\text{tot}}(\mathbf{o}, \mathbf{a})]\right| \leq L_Q\, W_2(\pi_w(\mathbf{o}), \pi_\phi(\mathbf{o})),$$

where the inequality follows from the dual formulation of Lipschitz functions and Wasserstein distances. Finally, applying Proposition 4.2 gives

$$\left|\mathbb{E}_{\mathbf{a} \sim \pi_w(\mathbf{o})}[Q_{\text{tot}}(\mathbf{o}, \mathbf{a})] - \mathbb{E}_{\mathbf{a} \sim \pi_\phi(\mathbf{o})}[Q_{\text{tot}}(\mathbf{o}, \mathbf{a})]\right| \leq L_Q\left(\mathbb{E}_{\mathbf{z} \sim p_0}\|\mu_w(\mathbf{o}, \mathbf{z}) - \mu_\phi(\mathbf{o}, \mathbf{z})\|_2^2\right)^{1/2}.$$

The proof is completed.  □

# E   TRAINING DETAILS OF MAC-FLOW

This section describes the implementation details of MAC-Flow.

**Network architectures.** MAC-Flow is implemented on multi-layer perceptrons (MLPs) with hidden sizes $[512, 512, 512, 512]$ for all networks, including the joint flow policy, the critics, and the factorized one-step policies. Layer normalization is applied consistently to further improve stability.

**Flow matching.** As described in Section 4.2, we adopt the simplest flow-matching objective (Equation 6) based on linear interpolation and uniform time sampling. For all environments, we use the Euler method with a step count of 10 to approximate the underlying ODE dynamics (Algorithm 2). This ensures that the joint flow policy $\mu_\theta(o, z)$ captures the multi-modal structure of coordinated behaviors while remaining computationally efficient at training and inference.

**Value learning.** We train individual critics $\{Q_{\theta_i}\}_{i=1}^I$ under the IGM principle with TD error update. We basically follow the $\mathrm{mean}(Q_1, Q_2)$ method, instead of $\min(Q_1, Q_2)$, to avoid pessimism in an offline RL setting. The one-step policies are optimized to maximize the global $Q$-function while simultaneously minimizing the distillation loss between the flow-based joint policy and the factorized policies. To calculate global $Q$, we use the average mixer for each agent's $Q$ value (Sunehag et al., 2017; Danino & Shimkin, 2025). To practically enforce the Lipschitz constraint required in Proposition 4.3, we apply *layer normalization* (Ba et al., 2016; Park et al., 2022) to all critic networks, which we found to be crucial for stabilizing value learning in multi-agent coordination.

**One-step policy learning.** After training the flow-based joint policy $\mu_\theta(o, z)$, we factorize it into decentralized one-step policies $\{\mu_{w_i}(o_i, z_i)\}_{i=1}^I$. This stage jointly optimizes two objectives in Equation 9: *(i)* Q-maximization and *(ii)* BC distillation. In general, we set the BC distillation coefficient $\alpha = 3.0$ as the default. In practice, we alternate between updating the critics and the one-step policies, using the same batch of transitions (Algorithm 1).

**Flow matching and policy for discrete action space.** For SMACv1 and SMACv2, which are discrete action control tasks, we model actions as one-hot vectors and learn a continuous vector field over the simplex. Specifically, for each transition, we form a linear path from Gaussian noise $x_0 \sim \mathcal{N}(0, I)$ to the one-hot actions $x_1 = \mathrm{onehot}(a)$, sample $t \sim \mathrm{Unif}([0, 1])$, set $x_t = (1-t)x_0 + tx_1$, and supervise the BC flow field $v_\phi(o, x_t, t)$ with the target velocity $x_1 - x_0$ via an MSE loss. To obtain a one-step sampling policy, we distill the multi-step Euler integration of $v_\phi$ into a single-step flow head that outputs logits over actions. During actor updates, we add a $Q$-guidance term that maximizes the mixed value of the actions proposed by the one-step policy, with an optional normalization of the guidance magnitude. At the training phase, target actions for TD backups are sampled from the flow policy; at deployment, we use $\arg\max$ with temperature control value, and optionally apply legal-action masking before the softmax.

**Online fine-tuning.** For the offline-to-online experiments (RQ3), we do not consider *symmetric sampling*, which reuses the offline dataset during online training (Ross & Bagnell, 2012), unlike prior research (Ball et al., 2023; Eom et al., 2024; Lee & Kwon, 2025). Instead, the agent is trained purely on newly collected online rollouts for an additional $500K$ gradient steps, continuing from the offline pretraining checkpoint.

**Training and evaluation.** We train MAC-Flow with 1M gradient steps for SMACv1 and SMACv2, and $500K$ steps for MPE and MA-MuJoCo. For offline-to-online training, we first perform $500K$ steps of offline training, followed by $500K$ steps of online training. We evaluate the learned policy every $50K$ steps using 10 evaluation episodes. For the main results in Tables 1 and 2, we report average performance and two standard deviations across 6 random seeds for the table and tolerance interval for the learning curves.

# F   IMPLEMENTATION DETAILS

The main objective of this work is to alleviate the gap between performance and inference speed in a multi-agent setting. Therefore, we deliberately adopt a simple network architecture, such as a multi-layer perceptron (MLP), rather than resorting to more complex or specialized designs. The simplicity of MLPs provides several advantages: *(i)* they allow for faster inference and lower computational overhead, which is critical for scalability in multi-agent settings; *(ii)* they facilitate stable training and clear evaluation of the proposed algorithmic contribution without the confounding effects of intricate architectures; and *(iii)* they serve as a neutral baseline architecture, demonstrating that the observed improvements stem from our framework itself, not from architectural sophistication.

## F.1   BASELINE ALGORITHMS

**BC**. This is a simple behavioral cloning (also known as imitation learning) algorithm. For the continuous action domain, we design it as a Gaussian policy with a unit standard deviation. For the discrete action domain, we parameterize the policy as a categorical distribution, where the policy network outputs unnormalized logits over all possible actions and the resulting action probabilities are obtained via a softmax function. Our network scheme is $[512, 512, 512, 512]-$sized MLPs, which is also our default network architecture, for all environments.

**Diffusion BC**. This is a diffusion-based extension of `BC`. Instead of directly regressing expert actions, `Diffusion BC` learns a denoising diffusion process: given an observation and a noisy version of the expert action, the policy predicts the noise to recover the clean action. For the continuous action domain, we model actions as Gaussian vectors, where the training objective is to predict Gaussian noise added during the forward diffusion process. At inference, the policy generates actions by reverse diffusion conditioned on current observations, starting from Gaussian noise. For the discrete action domain, we represent actions as one-hot vectors and apply diffusion in the relaxed continuous space. For both domains, we use $[512, 512, 512, 512]$-sized MLPs, augmented with a sinusoidal timestep embedding. We consider a Gaussian diffusion scheduler with 200 timesteps to govern the forward and reverse processes for all environments.

**Flow BC**. This is implemented on top of the same codebase as `MAC-Flow`, sharing the same flow-matching implementation. However, `Flow BC` does not consider Q maximization and distillation; in other words, it directly uses a trained full flow policy in training and deployment. We train individual vector fields for individual BC flow policies to distribute them into decentralized setups. For discrete control, actions are represented as one-hot vectors and flow matching is applied between Gaussian noise and the one-hot target along a linear path, with the policy trained to predict the corresponding velocity field. We consider 10 flow steps and $[512, 512, 512, 512]$-sized MLPs for all environments.

**MATD3BC** (Fujimoto & Gu, 2021). We reimplement it on our codebase by referring to the official open-source implementation of `TD3BC`. This is a multi-agent extension of TD3 with an additional BC regularization term. The BC parameter $\alpha$ is set as 2.5 for all environments. We train two critic networks via clipped double Q-learning, and target critic networks via the Polyak averaging with $\tau = 0.005$ (Polyak & Juditsky, 1992). For multi-agent settings, we employ the mixer for Q-value networks. The policy and value networks are parameterized as $[512, 512, 512, 512]$-sized MLPs.

**MABCQ** (Fujimoto et al., 2019). We reimplement it on our codebase by referring to the official open-source implementation of `BCQ`. We employ the mixer for Q-value networks. For discrete control, we design the policy as a softmax policy network with twin Q-networks. It masks out actions whose probability falls below a BC threshold $\alpha = 0.4$ and selects the maximum Q-value only among admissible actions. The critic is trained with TD targets using the masked actions set, and the actor is updated via cross-entropy loss against the action of the replay buffer. We use $[512, 512, 512, 512]$-sized MLPs for both actor and critic, with a target update period of 200 steps.

**MACQL** (Kumar et al., 2020). We reimplement it on our codebase by referring to the official open-source implementation of `CQL`. The critic consists of twin Q-networks with TD loss and a conservative penalty that lowers Q-values on OOD actions via a log-sum-exp term. Additionally, we employ the mixer for Q-value networks to consider multi-agent coordination. For discrete controls, the policy is set as a categorical distribution, and the illegal actions are masked during selection; for continuous controls, we use the Gaussian policy. We use $[512, 512, 512, 512]$-sized MLPs for both actor and critic, a target update period of 200 and every iterations ($\tau = 1.0$ for discrete domains) and

($\tau = 0.005$ for continuous domains), conservative weight 3.0, and 10 sampled actions per state for calculating the conservative loss for all environments.

**OMAR** (Pan et al., 2022). We reimplement it on our codebase by referring to the official open-source implementation of `OMAR`. This algorithm employs `CQL`-style conservative critic regularizer, a cross-entropy method (CEM)-based policy improvement head. The policy collects candidate actions via iterative CEM and is trained to imitate the best-Q candidate and maximize Q via a small L2 penalty. We use $[512, 512, 512, 512]$-sized MLPs and set the hyperparameter as follows: target update rate $\tau = 0.005$, CQL regularizer (10 OOD samples, and $\alpha = 3.0$), and CEM process (3 iterations, 10 samples, 10 elites per step, and mixing coefficient 0.7).

**OMIGA** (Wang et al., 2023). We reimplement it on our codebase by referring to the official open-source implementation of `OMIGA`. The critics use twin $Q$-networks combined by a learnable state-dependent mixing network, while a separate $V$-network provides a baseline for variance reduction. The policy is updated with advantage-weighted regression (Peng et al., 2019). Target networks for $Q$, $V$, and the mixer are updated via Polyak averaging with $\tau = 0.005$. We use $[512, 512, 512, 512]$-sized MLPs for all networks, a mixer embedding dimension of 128, advantage scaling coefficient $\alpha = 10.0$, gradient clipping at 1.0, and weight clipping for policy updates at 100.0.

**MADiff** (Zhu et al., 2024). We use the official open-source implementation of `MADiff`. This repository provides two variants: `MADiff-C` and `MADiff-D` for centralized and decentralized versions, respectively. Given that our problem formulation is a Dec-POMDP, we select decentralized variants, that is, `MADiff-D`. We train a conditional diffusion policy on joint demonstration trajectories with centralized data, then deploy it with decentralized execution. Each agent at deployment conditions only on its own local observation and optional history and samples its action via reverse diffusion while jointly predicting teammates' trajectories. We keep the same architecture and hyperparameters in the original paper and set the diffusion scheduler to Gaussian DDPM with 200 denoising steps for all environments.

**DoF** (Li et al., 2025a). We use the official open-source implementation of `DoF`. We use the `DoF-Trajectory` agent with the `W-concat` factorization. Training and deployment settings are identical to the original paper. For sampling, we use a Gaussian DDPM scheduler with 200 denoising steps across all environments.

### F.2 GIT REPOSITORY FOR BASELINE IMPLEMENTATION

The implementation adheres closely to the aforementioned official code as follows.

- TD3BC: https://github.com/sfujim/TD3_BC
- BCQ: https://github.com/sfujim/BCQ
- CQL: https://github.com/aviralkumar2907/CQL
- OMAR: https://github.com/ling-pan/OMAR
- OMIGA: https://github.com/ZhengYinan-AIR/OMIGA
- MADiff: https://github.com/zbzhu99/madiff
- DoF: https://github.com/xmu-rl-3dv/DoF/tree/main

# G EXPERIMENTAL DETAILS

## G.1 SMACv1 AND SMACv2

**SMACv1**, introduced by Samvelyan et al. (2019), serves as a prominent benchmark for assessing cooperative MARL algorithms. Built on the StarCraft II real-time strategy game, SMAC simulates decentralized management scenarios where two opposing teams engage in combat scenarios, with one team controlled by built-in AI and the other by learned multi-agent policies. Agents receive partial observations restricted to a local sight range (*e.g.*, nearby allied and enemy units, distances, health, and cooldowns), while the complete state information is available only for centralized learning of some algorithms. The discrete action space includes moving in four directions, attacking visible enemies within range, stopping, and a no-op, which makes coordination strategies such as focus fire, kiting, and terrain exploitation critical for success. The testbed defines a suite of combat maps of varying difficulty (*e.g.*, homogeneous battles like 3m and heterogeneous battles such as 2s3z), enabling systematic evaluation of algorithms across easy, hard, and super-hard scenarios. Reward signals are shaped by combat outcomes, including damage dealt, enemy kills, and victory, making SMACv1 a rigorous benchmark for addressing key MARL challenges such as credit assignment, cooperation under partial observability, and scalability to larger teams. Our evaluation focuses on five SMAC maps: 3m, 8m, 2s3z, 5m_vs_6m, and 2c_vs_64zg.

**SMACv2** extends the original SMAC benchmark to provide a more robust and challenging testbed for MARL. Unlike SMACv1, where difficulty mainly arises from heterogeneous unit compositions and map layouts, SMACv2 introduces environment stochasticity and increased diversity in scenarios to better approximate real-world complexity. In particular, unit placements, initial health, and enemy strategies are randomized across episodes, requiring policies that generalize beyond fixed configurations. The benchmark also rebalances reward signals to reduce overfitting to deterministic strategies and to encourage learning more adaptive coordination behaviors. By encompassing a broader range of maps and stochastic battle conditions, SMACv2 provides a more rigorous evaluation of MARL algorithms in terms of robustness, generalization, and sample efficiency. Our evaluation focuses on three SMAC maps: terran_5_vs_5, terran_10_vs_10, and zerg_5_vs_5.

**SMAC datasets.** Experiments utilized datasets from the off-the-grid offline dataset (Formanek et al., 2023), which provides offline trajectories for SMACv1 and SMACv2. The benchmark contains a variety of trajectories generated by different policies with varying quality, *including* Good, Medium, and Random behaviors, thereby covering a wide performance spectrum (whereas there is only the Replay dataset for SMACv2). Each dataset consists of observations, actions, legal action information, and rewards, following the decentralized agent structure of SMAC.

## G.2 MA-MUJOCO

**MA-MuJoCo** is a continuous-control benchmark that extends the classical MuJoCo locomotion suite to multi-agent settings (Peng et al., 2021). In MA-MuJoCo, a single robot, *e.g.*, Halfcheetah, Hopper, and Ant, is decomposed into multiple controllable parts, each assigned to an individual learning agent. Each agent receives local observations corresponding to its controlled joints and must produce continuous actions to coordinate with others for effective global locomotion. Rewards are typically shared among all agents based on the overall task performance, creating a cooperative continuous-action control problem. MA-MuJoCo focuses on fine-grained coordination in high-dimensional continuous dynamics, making it complementary for evaluating scalability and cooperation in MARL.

**Dataset.** For offline MARL, we employ the dataset collected by Wang et al. (2023). The dataset provides trajectories collected under a variety of policies with different quality levels, *including* Expert, Medium, Medium-Expert, and Medium-Replay, thus covering a wide spectrum of data distributions for 6-Halfcheetah, 3-Hopper, and 2-Ant. Heading number is the number of agents.

## G.3 MPE

**MPE**, originally introduced by Lowe et al. (2017) as a suite of simple particle-based worlds with continuous observation-action space and basic simulated physics, serves as a foundational benchmark for cooperative and competitive MARL tasks. It features two-dimensional scenarios where agents can move, interact, communicate, and observe each other within a partially observable setting. This

emphasizes coordination, communication, and emergent behaviors in multi-agent settings. Common scenarios include Cooperative Navigation (also known as Spread), where agents must cover landmarks while avoiding collisions; Predator-Prey, where predator agents pursue and capture prey agents; and World tasks, which involve more complex interactions like gathering resources or navigating with environmental elements. These environments are widely adopted for their scalability, ease of customization, and ability to test algorithms on communication-oriented problems. Performance in MPE is typically normalized using the following equation: $100 \times (S - S_{\text{random}})/(S_{\text{expert}} - S_{\text{random}})$, where $S$ is the score of the evaluated policy, $S_{\text{random}}$ is the performance from a random policy (159.8), and $S_{\text{expert}}$ is the performance of an expert-level policy (516.8).

**Dataset and Scenario Selection.** In our experiments, we focused solely on the Spread scenario dataset (*including* Expert, Medium, Medium-Replay, and Random), as implemented in a JAX-based framework to ensure efficient computation and compatibility with modern acceleration tools. This choice was necessitated by challenges in accessing the Predator-Prey (PP) and World (WD) datasets, despite their use in prior works by Pan et al. (2022) for offline MARL. Furthermore, adapting customized environments for PP and WD proved difficult due to the requirement of loading pre-trained policies, compounded by limited documentation on integration processes, which hindered compatibility with widely used pre-trained models (Formanek et al., 2024).

## G.4 HYPERPARAMETERS

| Hyperparameter | Value |
| --- | --- |
| Gradient steps | $10^6$ (SMACv1 and SMACv2), $2 \times 10^5$ (MA-MuJoCo and MPE) |
| Batch size | 64 |
| Flow step | 10 |
| BC coefficient | 3.0 |
| Network configuration | $[512, 512, 512, 512]$ |
| Polyak averaging coefficient | 0.005 |
| Discount factor | 0.995 |
| Optimizer epsilon | $10^{-5}$ |
| Weight decay | 0 |
| Policy learning rate | $3 \times 10^{-4}$ |
| Value learning rate | $3 \times 10^{-4}$ |
| Layer normalization | True |
| Optimizer | Adam |

# H    ADDITIONAL RESULTS

This section provides supplementary analyses that further complement the main results presented in the paper. Specifically, we first include several additional ablation studies that examine the robustness of our framework under varying design choices, such as alternative policy parameterizations and critic configurations. Second, we check the training time of our approach and baselines. Third, we report the full learning curves of `MAC-Flow` corresponding to the performance tables in the main text.

## H.1    ABLATION STUDY

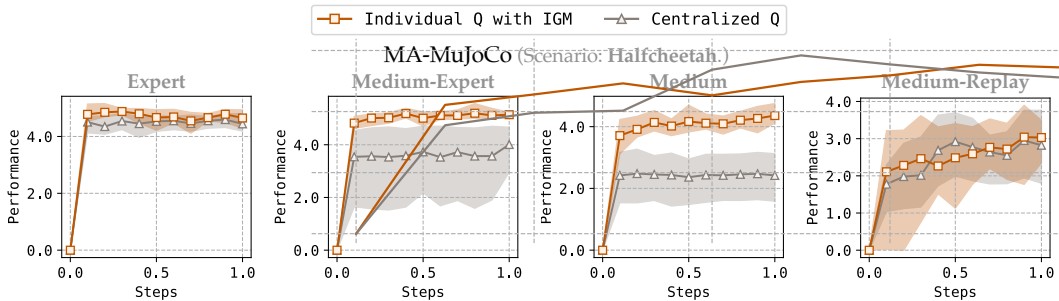

Figure 8: **Ablation on critic configuration.** This ablation shows the differences between individual Q training under IGM and centralized Q training. The reported point and shaded area represent the average and tolerance interval from 6 random seeds.

**Individual $Q$ with IGM vs. Centralized $Q$.** In Figure 8, across all MA-MuJoCo datasets, the individual $Q$s based on IGM consistently outperform the centralized counterpart $Q$. While the centralized variants often suffer from lower stability and suboptimal convergence, the individual $Q$ based on IGM achieves higher performance and maintains table learning curves. This demonstrates that the IGM formulation enables reliable value estimation in multi-agent settings by preserving individual-global consistency, which is critical for cooperative policy learning. We conjecture that such empirical observations originate from a representation bottleneck of simple MLP networks. Specifically, the centralized critic should extract coordination patterns from the joint observation-action space. In practice, shallow MLP architectures struggle to capture such high-order dependencies, leading to severe information compression and a bottleneck in representing coordination. To sum up, the limitations observed in centralized critics likely stem not from the principle of centralization itself, but from the restricted capacity of MLP-based function approximators when faced with large joint observation-action spaces.

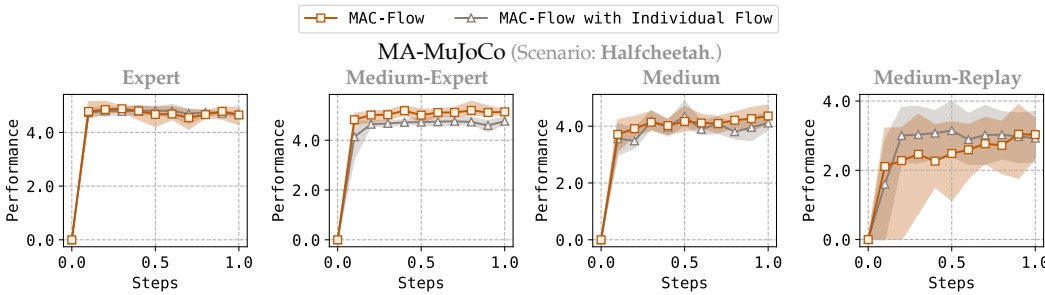

Figure 9: **Ablation on policy type in Stage I.** This ablation shows the differences between individual flow policies and joint flow policy for stage I. The reported point and shaded area represent the average and tolerance interval from 6 random seeds.

**Individual BC Flow vs. Joint BC Flow (Stage I).** Figure 9 investigates the effect of policy configuration in Stage I by comparing individual flow policies against our joint flow policy. Across all MA-MuJoCo datasets, our basic configuration and its variant show stable and powerful performance.

Interestingly, unlike the centralized $Q$ ablation, the joint flow policy does not entirely collapse. We attribute this difference to the expressive capacity of flow-matching methods, which are specially designed to approximate complex distributions. Whereas centralized critics implemented with shallow MLPs face a severe representation bottleneck when mapping from joint observation-action inputs, flow-based policies retain stronger inductive biases for capturing distributional structure. This expressiveness enables joint flows to remain viable in principle, though their optimization often remains more challenging than individual flows. Comprehensively, given the trade-offs, we opt for the joint policy to enhance sample efficiency, reduce memory usage (with respect to network load), and facilitate faster training.

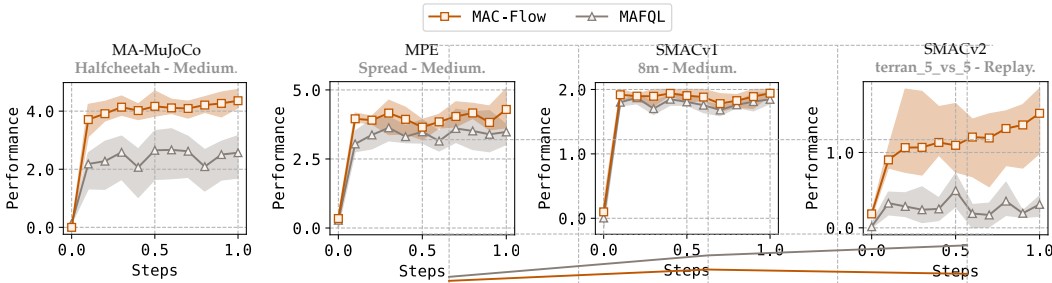

Figure 10: **Performance comparison with MA-FQL.** We compare our solution and the naive extension of the FQL across four benchmarks. The reported point and shaded area represent the average and tolerance interval from 6 random seeds.

**MAC-Flow vs. MA-FQL.** Figure 10 compares the performance between our proposed solution and the multi-agent extension of FQL (Park et al., 2025), which is a close connection with our work. Comparing MAC-Flow against MA-FQL highlights that the transition from a single-agent to a multi-agent system is not trivial. In MA-MuJoCo, MPE, and SMAC benchmarks, MAC-Flow substantially outperforms MA-FQL in both convergence speed and final performance. In contrast, MA-FQL either stagnates at suboptimal levels or exhibits unstable progress, particularly in more complex environments. Note that we can simply put that MA-FQL is a fully decentralized version of MAC-Flow, especially in decentralized $Q$ without IGM and flow decentralized policy.

## H.2 Training Time

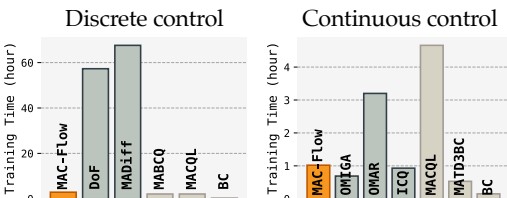

Figure 11: **Wall-clock time for training.** Reported numbers are measured on average of all tasks, which are included in discrete or continuous benchmarks.

We report the wall-clock training time for both discrete and continuous control benchmarks in Figure 11. MAC-Flow achieves substantially lower training time compared to diffusion-based methods such as DoF and MADiff. In discrete control, MAC-Flow trains nearly an order of magnitude faster than MADiff, while in continuous control, it also outperforms strong baselines like MA-CQL or OMAR, which require many bootstrapping steps. These results confirm that the one-step flow formulation of MAC-Flow yields not only efficient inference but also significantly reduced training cost. Note that CQL is implemented with independent $Q$ learning (De Witt et al., 2020) in discrete control, unlike continuous control.

## H.3  LEARNING CURVES OF MAC-FLOW

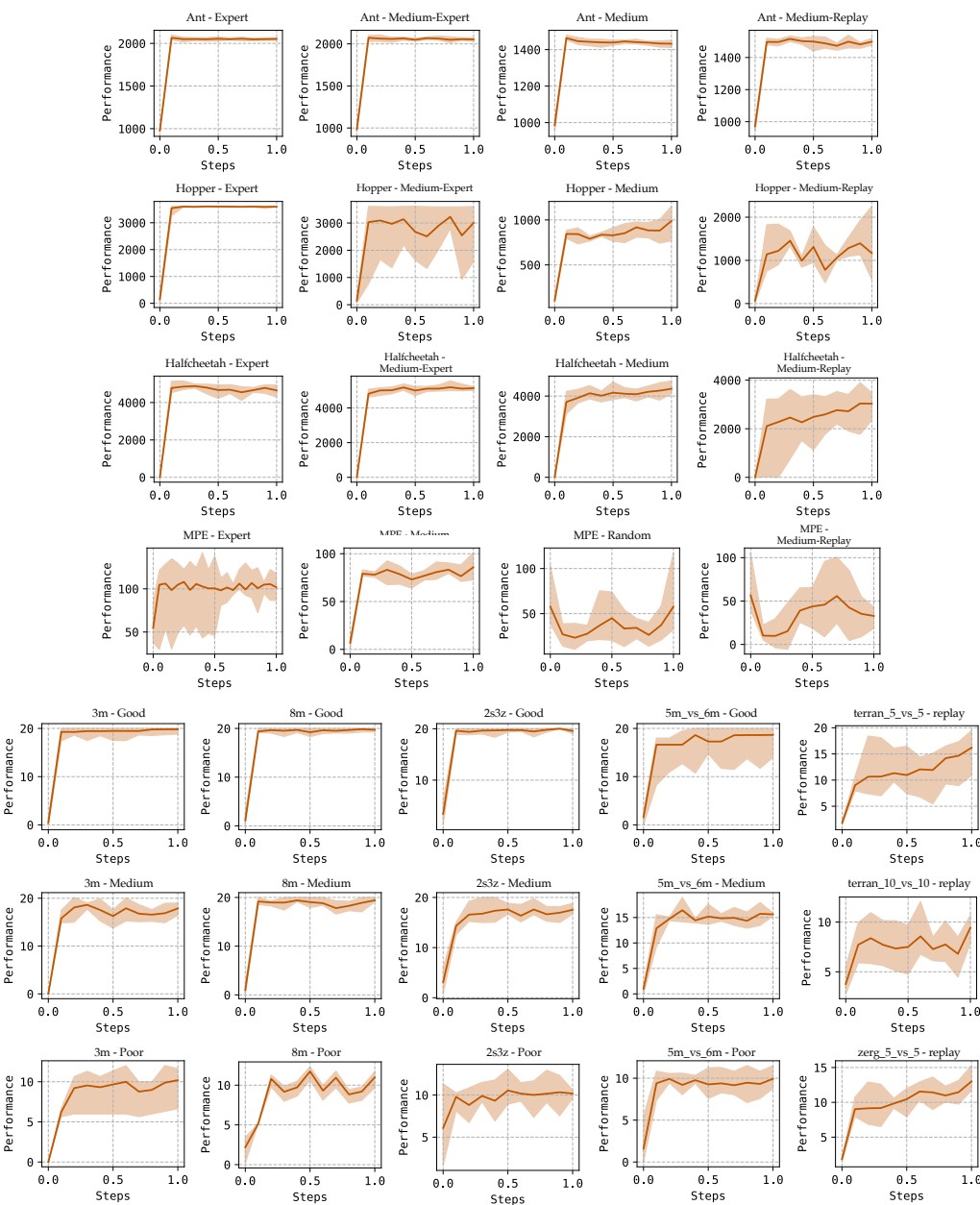

Figure 12: **Full learning curves for MAC-Flow.**

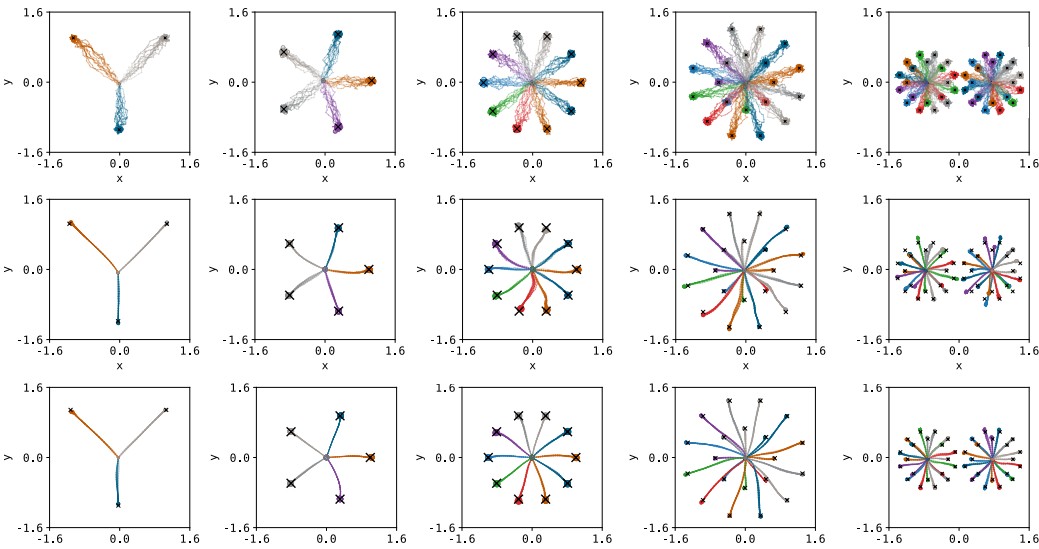

Figure 13: **Landmark covering game visualization.** (*Top*) Trajectories sampled from the dataset. (*Middle*) Trajectories sampled from the flow joint policy. (*Bottom*) Trajectories sampled from the individual policy.

### H.4 LANDMARK COVERING GAME: SCALING UP THE NUMBER OF AGENTS

To further examine how `MAC-Flow` scales with the number of agents, we extend the landmark-covering game (used in Section 4.3) up to $40$ agents from three agents. We generate an offline dataset consisting of $50$ trajectories, and train both the joint flow model and the distilled one-step policies for $1000$ iterations with a $64$ batch size.

Figure 13 visualizes the sampled trajectories from the dataset, flow joint policy, and individual policies, respectively. Up to $40$ agents, the flow joint policy successfully captures the multi-agent's joint action distribution and the appropriate partitioning of agents across landmarks ($\times$ mark). Although the flow model occasionally produces slightly dispersed trajectories, given our training setup, this could be mitigated by scaling up the hyperparameter. Next, the individual policies learn clean trajectories, since each agent only needs to reproduce its own trajectories, which is similar to expert demonstrations.

**Summary.** This scaling analysis confirms that `MAC-Flow` continues to capture stable coordination up to $40 agents$, with this model capturing joint structure. The results indicate that `MAC-Flow` can retain its effectiveness (according to flow policy) even as multi-agent dimensionality grows.

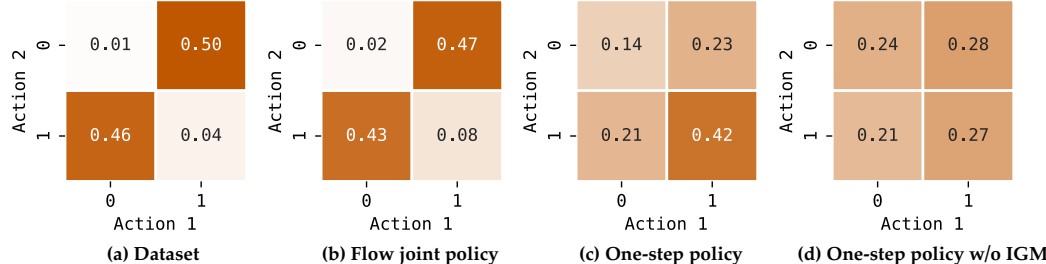

Figure 14: **Policy comparison in pure coordination game.** (*a*) Dataset distribution. (*b*) Distribution of flow joint policy. (*c*) Distribution of one-step policy with IGM. (*d*) Distribution of one-step policy without IGM.

### H.5 PURE COORDINATION GAME: BETTER POINT THAN BC GENERATIVE MODELING

This section presents a setting where IGM offers a clear advantage over BC-style generative modeling. This experiment isolates a simple yet revealing coordination structure where value information plays a decisive role in recovering the optimal joint behavior, whereas BC alone fundamentally fails due to dataset imbalance.

**A minimal pure coordination game.** We consider a two-agent binary-action environment with a unique high-value coordinated action, $(1, 1)$, yielding reward $+2$. All partially coordinated actions, $(1, 0)$ and $(0, 1)$, give a lower reward $+1$, while $(0, 0)$ yields zero. Critically, the offline dataset is intentionally biased, that is, it consists almost of the two asymmetric suboptimal modes $(0, 1)$ and $(1, 0)$, with only a small fraction of rare samples of $(1, 1)$ and $(0, 0)$. This generates a combinatorial ambiguity that challenges BC-style modeling, which must extrapolate optimal coordination from extremely weak statistical evidence (Figure 14 (*a*)).

**Flow joint policy vs. `MAC-Flow` vs. Factored policy without IGM.** In this setup highlights a common failure pattern in BC generative modeling (Flow joint policy). Because BC methods optimize likelihood without considering value structure, the learned joint flow simply reproduces the dominant dataset frequencies, faithfully capturing $(0, 1)$ and $(1, 0)$ while ignoring the much rarer yet more valuable $(1, 1)$ mode. Even though this is expressive enough to represent the correct distribution, its objective provides no incentive to amplify the crucial but low-frequency coordination (Figure 14 (*b*)).

In contrast, introducing IGM during distillation (MAC-Flow) substantially changes the learning dynamics. When the joint flow is mapped into factorized per-agent policies through value-guided IGM constraints, each agent receives a consistent signal that selecting the action 1 is slightly more aligned with high-value coordination, even though this is rarefied in the dataset. The IGM mechanism exploits these weak cues by forcing individual policies to agree on action choices that maximize the centralized value function, effectively amplifying faint optimality information that pure BC cannot utilize. As a result, the distilled factorized policy concentrates significantly more probability mass on the optimal $(1, 1)$ configuration (Figure 14 (*c*)).

Whereas the version without IGM collapses toward an almost uniform product distribution that fails to represent any coordinated behavior. In this case, each agent's marginal policy distributes probability mass nearly evenly over its two actions, causing the joint distribution to degenerate into four nearly equal-probability modes (Figure 14 (*d*)).

**Summary.** This example shows when IGM is strictly better than pure BC modeling. In settings where demonstrations contain only weak hints of coordinated optimal behavior, the IGM mechanism reliably extracts and amplifies value-relevant structure that generative models alone cannot recover. This controlled setup thus clarifies why MAC-Flow's value-guided distillation often outperforms BC-based generative modeling in more complex multi-agent domains.

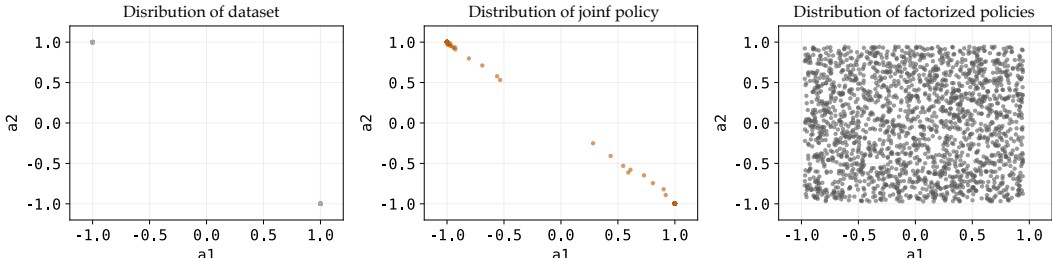

Figure 15: **XOR toy environment**. (*Left*)The dataset contains two anti-aligned modes. (*Center*) The joint flow recovers this non-factorizable structure. (*Right*) Factorized policies collapse into an almost uniform product distribution.

## H.6   XOR STRESS TEST: FAILURE MODE ANALYSIS

Although we provide theoretical justification for our bound, this subsection presents empirical counterexamples for transparency and trust, demonstrating how the method behaves in environments that intentionally violate the assumptions underlying the theory.

**When does IGM break?** We analyze settings where the IGM factorization is provably invalid. While `MAC-Flow` performs well in typical cooperative tasks, separability could fail under strong inter-agent coupling. To build trust in our theoretical claims, we present a controlled stress test where factorization is guaranteed to break.

**XOR coordination examples.** We introduce a minimal two-agent continuous XOR task in which the optimal joint actions are anti-aligned, *e.g.*, $(-1, +1)$ and $(+1, -1)$. Because the reward depends entirely on the relative actions of the agents, no per-agent Q-function can recover the optimal strategy independently, making IGM mathematically impossible in this domain. This setting provides a clean, analyzable failure case for joint-to-factorized policy distillation.

**Empirical counterexamples.** We train the `MAC-Flow` joint policy on an offline dataset exhibiting two sharp, disconnected high-density XOR modes. The learned flow successfully reconstructs these modes and captures their fundamentally non-separable geometry, confirming that the flow-matching stage is expressive enough to model complex, multi-modal joint behaviors. However, when this joint flow is distilled into per-agent policies, the factorized representation collapses into a near-independent product distribution, failing to reproduce the anti-aligned high-density regions and instead placing probability mass around the center of the action space. This degradation arises despite the joint model being correct, indicating that the failure is due to the intrinsic non-separability of the coordination structure rather than optimization or training artifacts.

**Summary.** The XOR stress test provides exactly the concrete failure mode. In this environment, the optimal joint action is intrinsically anti-aligned, causing the assumptions behind IGM factorization to break. As a result, value bounds cease to be predictive because the global optimum cannot be decomposed into per-agent optima. Consequently, `MAC-Flow` underperforms not due to model capacity or training issues, but because the coordination structure is fundamentally non-separable. This controlled setting, therefore, offers a transparent counterexample.

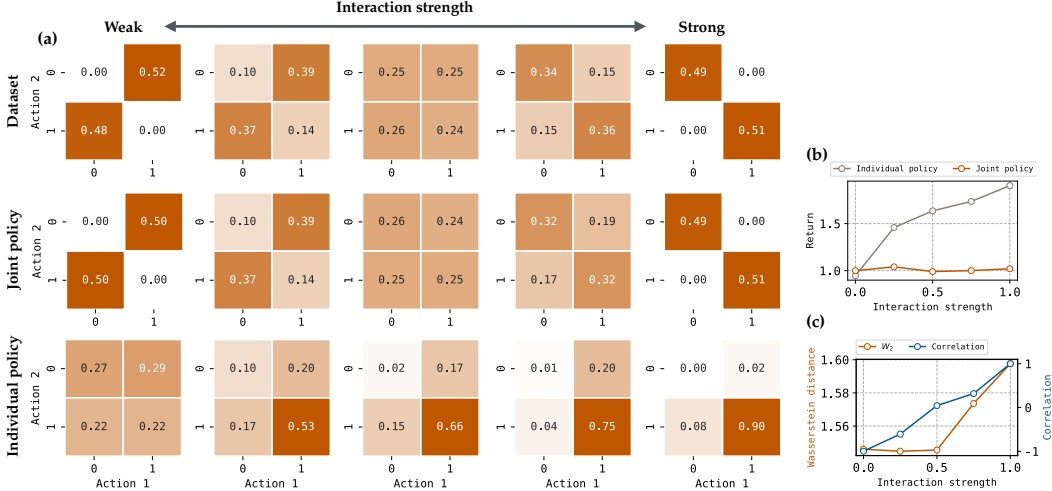

Figure 16: **Payoff game according to interaction strength.** (*a*) Sampled distributions.(*b*) Return differences. (*c*) Wasserstein distance between joint and individual policies. Correlation between the dataset and joint policy.

## H.7 PAYOFF GAME: ANALYSIS ACCORDING TO INTERACTION STRENGTH

To validate the value gap and $W_2$, we design a pay-off game according to the interaction strength, as an extension of the pure coordination game in Appendix H.5. The MDP setup is the same as the pure coordination game: each agent can select an action in $\{0, 1\}$, and the team reward is defined as $(0, 0) = 0, (1, 0)$ or $(0, 1) = 1, (1, 1) = 2$.

**How do we define interaction?** In this example, we define *interaction strength* as the degree to which one agent's optimal action depends on the other agent's action, or equivalently, how far the joint action distribution deviates from a factorized form $p(a^1, a^2) = p(a^1)p(a^2)$. Under this view, diagonal joint actions $(0, 0)$ and $(1, 1)$ represent strong interaction, because the optimal behavior requires perfectly matching actions and thus induces full coupling. In contrast, off-diagonal actions $(1, 0)$ and $(0, 1)$ correspond to weak interaction, where each agent's marginal behavior remains nearly independent. To probe this, we generate datasets by mixing diagonal and off-diagonal modes with a controllable interaction parameter $\zeta \in [0, 1]$.

$$\mathcal{D}^\zeta = \zeta \times \{(0, 0), (1, 1)\} + (1 - \zeta) \times \{(0, 1), (1, 0)\}$$

Herein, $\alpha = 0$ yields purely weak interaction, and $\alpha = 1$ yields purely strong interaction.

**Results.** Figure 16 compares the flow joint policy and individual policies trained under increasing interaction strengths ($\{0.00, 0.25, 0.50, 0.75, 1.00\}$) using the sampled joint-action matrices, the return, and the corresponding Wasserstein distance $W_2$. In Figure 16 (*a*), across all interaction levels, the flow joint model reproduces the offline dataset distribution. It means that the BC flow learning is effectively capturing the observed joint-action modes. On the other hand, the individual policy shows a different pattern driven by the IGM-based Q maximization rather than simply BC. As interaction increases, the IGM objective encourages the individual policies to concentrate probability mass on the optimal joint actions rather than merely reproducing the empirical frequencies.

To quantify its differences, Figure 16 (*b*) clarifies how interaction strength influences the learned policies. The flow joint policy achieves the near-optimal reward of approximately 1.0, matching the reward level of the dataset since it directly models the full joint action distribution. In contrast, the return of the individual policy increases gradually as the interaction strength rises. This occurs because the number of samples in the optimal condition increases, thereby making the conditions for identifying coordinated behaviors more pronounced. Next, Figure 16 (*c*) confirms that $W_2$ rises with the interaction strength, reflecting the growing structural mismatch between the expressive joint flow and the restricted factored representation.

**Summary.** As interaction strength increases, the Wasserstein distance $W_2$ between the joint and individual policies grows, reflecting the structural mismatch that factorized policies cannot eliminate.

