# OpenReview forum: "Multi-agent Coordination via Flow Matching"
_ICLR.cc/2026/Conference — ICLR 2026 Poster_

### Official Review · Reviewer_Z5Fb · 2025-10-29

**Soundness:** 2
**Presentation:** 3
**Contribution:** 2
**Rating:** 4
**Confidence:** 4

**Summary:**

The paper proposes MAC-Flow, an offline MARL framework that (1) learns an expressive flow-matching–based joint policy via behavioral cloning, then (2) factorizes / distills it into decentralized one-step policies optimized with a behavioral-regularized actor–critic objective and the Individual-Global-Max (IGM) principle. The pitch is to retain diffusion-like expressiveness for multi-modal joint action distributions while achieving Gaussian-policy–like inference speed. Experiments span different dataset, reporting faster inference vs diffusion MARL at comparable performance, and similar speed to Gaussian baselines

**Strengths:**

1. The paper clearly identifies—and structures the solution around—the core trade-off in offline MARL: diffusion policies capture multi-modal joint behaviors but are slow, while Gaussian one-step policies are fast but brittle for coordination.

2. The paper connects to flow-matching literature and positions MAC-Flow as a MARL counterpart to single-agent flow-distillation and Flow Q-Learning, showing conceptual continuity with recent advances.

3. The two-stage design is guided by the Individual-Global-Max principle, aligning per-agent policies with the global optimum and providing a standard, scalable route to decentralized execution

3. On four benchmarks, MAC-Flow demonstrates that you can retain coordination performance while achieving faster inference than diffusion-based MARL—matching Gaussian-policy speed without its expressiveness limits.

**Weaknesses:**

1. The method references “mathematical guarantees” around joint-to-factorized policy learning with IGM, but the paper (as given) does not present a formal theorem / conditions under which the distilled one-step policies provably preserve the global optimum of the learned joint flow.

2. While the authors claim flow matching combines diffusion’s expressiveness and Gaussian’s speed, the paper lacks a deeper justification for why flow matching provides better inductive bias or coordination representation in MARL. There is no theoretical or empirical comparison against diffusion-based MARL such as Graph Diffusion for Robust Multi-Agent Coordination (2025).

3. The evaluation omits stress tests under partial observability, agent scaling, and dynamic teammate shifts—scenarios where flow-based policies might degrade.

**Questions:**

1. Under what assumptions does the factorized one-step policy set preserve the argmax consistency with the flow-learned joint policy? Any error decomposition (e.g., from flow fitting + distillation + value learning) that bounds sub-optimality?

2. Can you report MI(agent i; agent j | o) or diversity / mode-coverage metrics comparing joint flow vs distilled policies, and ablate IGM on/off?

3. How does the policy behave under partial-observability stress (masking % of features) or opponent distribution shift?

4. Provide cases where MAC-Flow under-performs Gaussian or diffusion policies—what patterns are not captured by the distilled factorization?

---

> ### Author Response · Authors · 2025-11-16
> **Response to reviewer's concerns (1/5)**
>
> We sincerely thank the reviewer for the time and effort dedicated to evaluating our submission. Your detailed comments and constructive questions are extremely valuable to us, and we truly appreciate your careful reading of both the technical ideas and empirical results. In the following response, we aim not only to address each of your questions point-by-point, but also to clarify a few aspects where we believe certain concerns may stem from incomplete or unintended interpretations of our method.
>
> ---
> > **W1** The method references “mathematical guarantees” around joint-to-factorized policy learning with IGM, but the paper (as given) does not present a formal theorem / conditions under which the distilled one-step policies provably preserve the global optimum of the learned joint flow.
>
> We appreciate the reviewer’s careful reading and insightful comments. We would like to clarify that the “mathematical guarantees" referenced in the paper refer to bounds on joint-to-factorized deviation, not a preservation theorem claiming equality of global optima.
> - Under Definition 4.1 and a shared base noise, Proposition 4.2 provides an upper bound on the joint–factorized discrepancy in terms of the square root of the $W_2$ distillation loss.
> - Assuming an L-Lipschitz $Q_{\text{tot}}$, Proposition 4.3 converts this discrepancy into a value-gap bound.
>
> Together, these results yield approximation guarantees under explicit assumptions, not a theorem ensuring that the decentralized policies preserve the global optimum of the centralized flow. Such exact preservation would require extremely strong, non-generic conditions in MARL, whereas bounded deviation is a meaningful guarantee.
>
> `Revision.` To avoid any ambiguity, we will revise the manuscript to explicitly state in Section 4.2 as follows.
> ```
> Note that these properties collectively characterize bounded performance degradation under explicit assumptions rather than perfectly global optimality preservation.
> ```
>
> ---
> > **W2-2**. Missing related work and comparison with Graph Diffusion for Robust Multi-Agent Coordination 2025
>
> We thank the reviewer for pointing out this relevant and interesting work. We will cite it in the revised Related Work section and include a brief discussion on how its goal and modeling approach complement our contribution.
>
> Regarding empirical comparison, we note two important limitations: The codebase implementation is currently unavailable. It makes a reproducible comparison infeasible. To the best of our knowledge, the evaluation protocol differs from standard SMAC-v1, where the maximum achievable score is $20$. The paper appears to adopt a different scoring method, and as a result, the numerical values are not directly comparable without access to the exact implementation details.
>
> Finally, we would like to emphasize that diffusion-based modeling and flow matching as orthogonal research dimensions conceptually. The cited work focuses on explicitly modeling inter-agent relations through graph-based diffusion, which is complementary to our objective. In contrast, MAC-Flow aims to achieve scalable coordination through the synergy between IGM-based factorized composition and efficient flow-based distillation, rather than introducing a new inter-agent relational architecture. Thus, we see the two approaches as addressing different aspects of the MARL design space.

---

> ### Author Response · Authors · 2025-11-16
> **Response to reviewer's concerns (2/5)**
>
> > **W2-1**. Deeper justification of design choices of MAC-flow
>
> We appreciate the reviewer’s thoughtful comment. We would like to clarify that our claim is not that flow matching itself is the sole or dominant source of framework’s performance. Rather, the central contribution of MAC-Flow lies in the synergy between IGM-based factorization and an efficient flow-based distillation mechanism.
>
> `Clarification.` We stated that “flow matching is comparable (not outperform) in expressiveness to previous diffusion solutions.” We never claim that flow matching exceeds diffusion in generative capacity. More precisely, diffusion and flows are both good expressive models, and our intention is not to rank them by expressiveness but to explain our design choice within our MARL framework. Empirically, in Table I of original paper, MAC-Flow performs comparably to DoF on average, but DoF tends to be slightly stronger on certain benchmarks. On the other hand, MAC-Flow achieves highly efficient inference compared to such diffusion methods.
>
> `Core contribution.` Flow matching is chosen not because it alone provides a superior inductive bias, but because it offers a consistent, and theoretically tractable distillation mechanism that integrates with IGM-based compositions. That is, MAC-Flow demonstrates that expressive joint modeling, IGM decomposition, and efficient flow-based distillation yields scalable coordination. This is not that flow matching alone is superior to diffusion.
>
> As noticed, this is evidenced by Figure 7, which demonstrates that flow matching alone is not the driving factor of our framework's performance. Instead, the effectiveness stems from how flow-based distillation complements the IGM-based compositional structure, allowing us to maintain expressiveness while achieving scalability.
>
> `Diffusion policies.` We would like to highlight an important distinction: generative RL approaches behave more like BC-based control policies rather than RL-driven policy optimization. In contrast, MAC-Flow is an RL framework, where the flow model learns joint behavior, and the final decentralized policies are optimized within an RL loop. This difference is central to our design and contributes to the improved scalability and coordination stability compared with BC-driven approaches.
>
> `Why flow matching and distillation?` To further address this question, we clarify the rationale behind our framework's design. We considered alternative approaches for the distillation component:
> - Diffusion training + Gaussian distillation: Training with diffusion models and distilling to Gaussian policies via IGM would create an inconsistent framework with increased complexity, reducing practical applicability.
> - Pure diffusion approaches: While expressive, these require extensive hyperparameter tuning and lack computational efficiency.
>
> We chose flow matching to achieve a unified framework with minimal tunable parameters. This design provides:
> Clear trade-off characterization between expressiveness and efficiency
> - Tractable theoretical bounds (W2 distance guarantees)
> - Practical implementation through a consistent, unified framework
> - Reduced hyperparameter burden compared to diffusion-based methods
>
> ---
> As an additional clarification, we also include a small toy example in the revision to further illustrate this point, although it is not central to our main experiments.
>
> `Pure coordination games.` In this example, two agents can choose binary actions among (0,0), (0,1), (1,0), and (1,1). (0,1) and (1,0), each providing a reward of +1. The truly optimal coordinated action (1,1) yields a higher reward (+2) but appears only very sparsely. the dataset is dominated by the two asymmetric modes ((0,1) and (1,0); only +1 rewards), while the truly high-value joint action appears only rarely ((1,1); get +2 rewards). It means that an individual agent can get a reward from selecting 0 or 1.
>
> `Results.`
> - The joint BC-style flow model simply reproduces the main dataset modes and fails to recover the rare coordinated mode.
> - Factored policies without IGM spread probability across all four modes, failing to extract weak coordination signals.
> With IGM enabled, the factorized policy aligns the agents’ action preferences and shifts most mass toward the optimal (1,1) mode.
> - The factorized policy with IGM more mass on the optimal joint action compared to without IGM, whereas the joint model continues to simply reflect the empirical dataset distribution. This confirms that IGM provides a useful form of value-guided coordination that cannot be obtained from generative modeling alone.
>
> `Summary.`
> This confirms that pure BC-based generative modeling sometimes fails in environments where the dataset contains weak evidence of the optimal coordinated behavior. We believe our contribution lies in providing a theoretically grounded yet practical framework, addressing known challenges of complexity and hyperparameter sensitivity in existing MARL methods [1-3].

---

> ### Author Response · Authors · 2025-11-16
> **Response to reviewer's concerns (3/5)**
>
> > **W3 & Q3.** (Robustness and analysis) Stress tests under partial observability, agent scaling, or dynamic teammate shifts.
>
> We thank the reviewer for raising the question of robustness under different coordination. We respond to each aspect below.
>
> `Partial observability.` We would like to clarify that **our focus is already on partially observable multi-agent settings**. All selected benchmarks (i.e., SMACv1 and v2, MPE, and MA-MuJoCo) are ​​standard Dec-POMDP environments with per-agent local observations rather than full state access. Thus, the reported results inherently evaluate MAC-Flow under partial observability. We will clarify this more explicitly in the problem formulation and experimental setup sections to avoid any confusion that our analysis is limited to fully observed settings.
>
> `Related to masking % of features.` We agree the importance of robustness task like  the suggestion of reviewer, feature-masking of observation, in the broader robustness literature (e.g., sensor corruption, adversarial observation dropouts, or environment perturbation). But we view them as orthogonal to our problem formulation. Our work focuses on cooperative offline Dec-POMDPs. Introducing masking would change the observation distribution beyond what is present in the offline dataset, thereby creating a different robustness setting that is typically studied as a separate research direction.
>
> `Agent scaling.` Our SMAC experiments already span a broad range of team sizes from small squads (3 agents in a 3m task) to larger teams with 10 agents (terran_10_vs_10 task). We would like to further clarify that our setting is offline MARL, where scaling the number of agents is intrinsically challenging: generating or collecting balanced joint datasets that remain sufficiently expressive across agent counts leads to severe combinatorial sparsity and distributional mismatch issues in offline RL. For this reason, large-scale agent-scaling stress tests are considerably more difficult to conduct in offline MARL than in online policy-learning settings.
>
> `Revision for agent scaling.` Even so, we fully agree that scalability is significant in the MARL domain. Therefore, **we newly ran additional experiments in landmark covering games where we consider a minimum of 3 agents to 40 agents in Appendix H.4**. These additional results show that MAC-Flow continues to maintain stable coordination behavior even as the number of agents grows substantially, highlighting that the proposed flow-based distillation remains effective under increasing multi-agent dimensionality.
>
> `Dynamic teammate shifts.` Dynamic teammate changes and ad-hoc coordination with previously unseen partners are indeed important robustness questions. However, they typically fall under ad-hoc teamwork and zero-shot coordination in (often online) MARL, which assumes changing teammates or novel partner policies during execution. In contrast, our work targets offline MARL with a fixed joint dataset and a fixed team composition, focusing on scalable joint-to-factorized policy learning under distributional shift in the data, rather than dynamic teammate adaptation.
>
> >**Q1**. Argmax consistency & error decomposition
>
> We thank the reviewer for the question. The factorized one-step policies preserve approximate argmax consistency with the flow-learned joint policy under the standard Individual–Global–Max (IGM) assumption. While this does not guarantee exact equality of optima, our framework provides a **bounded sub-optimality characterization**. Specifically, the value gap between the joint flow policy and the distilled factored policies can be decomposed into three sources of error:
>
> - Joint flow fitting error: deviation between the true optimal joint distribution and the learned joint flow model.
> - Distillation error: the $W_2$ discrepancy between the joint flow distribution and the factorized one-step policies induced through IGM-based matching.
> - Value learning error: approximation error from estimating $Q_{\text{tot}}$, which is needed to map distributional mismatch into a value gap.
>
> Combining these, Propositions 4.2-4.3 yield an explicit upper bound that converts the joint-factorized distributional deviation into a proportional bound on value loss (assuming a Lipschitz $Q_{\text{tot}})$. Thus, rather than claiming perfect optimality preservation, our guarantees establish bounded degradation under explicit and interpretable assumptions.

---

> ### Author Response · Authors · 2025-11-16
> **Response to reviewer's concerns (4/5)**
>
> > **Q2**. Mutual information(agent i; agent j | o), and IGM ablation
>
> We appreciate the reviewer’s suggestion to include mutual information $\operatorname{MI}$ analyses. In the revised manuscript, we complement our main results with a didactic toy experiment where these quantities can be directly measured and related to the theory.
>
> To address the reviewer’s concern regarding whether the theoretical link between small $W_2$ discrepancy and small value loss is empirically justified, we performed additional experiments in this didactic setup. This environment is intentionally designed so that the distributional geometry and value landscape can be precisely monitored, enabling a causal evaluation of the assumptions underlying Proposition 4.3.
>
> **Distributional proximity improves over training.** Figure 3(a) shows that the inter-agent mutual information steadily decreases, indicating that the joint and factored policies become increasingly aligned. This demonstrates that the distribution mismatch assumption is progressively better satisfied during learning.
>
> **Value gap decreases in tandem with distillation loss.** In Figure 3(b), the empirical value gap consistently tracks the distillation loss, supporting the theoretical prediction that a smaller discrepancy between joint and factored distributions leads to a proportionally smaller value deviation.
>
> **Empirical value gap remains below the Lipschitz bound.** Figure 3(c–d) shows that all checkpoints lie strictly below the theoretical Lipschitz envelope $L_QW_2$​, with a clear monotonic trend between $W_2$​ distances and value gaps. This provides direct empirical evidence that the Lipschitz smoothness of the learned $Q_{\text{tot}}$ is sufficiently well-behaved in practice.
>
> Overall, the measurements in Figure 3 validate that the assumptions underlying our theoretical relation are not only mathematically convenient but are also reasonably met during training, thereby strengthening the practical relevance of Proposition 4.3.
>
> ---
> > **Q4** Provide cases where MAC-Flow under-performs Gaussian or diffusion policies—what patterns are not captured by the distilled factorization?
>
> We appreciate the reviewer’s question. First of all, we would like to provide the theoretical insight into where MAC-Flow fails. The proposed solution would be degraded in environments where the underlying joint-action structure is fundamentally non-separable. For example, the IGM-based factorization cannot reproduce the correlations encoded in the joint policy, even if the joint flow is learned perfectly.
>
> More precisely, we would like to identify two representative patterns:
> - **High stochasticity cooperative tasks.** In SMACv2, the optimal joint behavior forms a broad, that is, the expressiveness becomes more and more important. In such cases, diffusion solvers could benefit from averaging over trajectories and can better represent this high-variance geometry. On the other hand, factorized one-step policies may struggle to extract a set of stable policies. This explains why MAC-Flow achieves lower performance compared to the DoF baseline.
> - **Non-separable coordination tasks.** Theoretically, our solution requires (i) a Lipschitz $Q_{\text{tot}}$​ and (ii) sufficiently small distributional mismatch, and these assumptions may not hold uniformly. To make this limitation explicit, we include a controlled didactic example, the XOR stress test, in the revised manuscript.
>
> **We include an additional concrete example in Appendix H.6** an analytically tractable XOR stress test that deliberately violates separability. In this environment, the optimal joint actions are anti-aligned and therefore provably non-factorizable, making any IGM-based decomposition mathematically incompatible with the true joint structure. Consistent with this theory, the learned joint flow successfully recovers the two disjoint high-density modes of the XOR dataset, whereas the distilled factorized policies collapse toward an almost uniform product distribution centered around the middle of the action space. This degradation provides a transparent illustration of the precise failure mode: when coordination requires intrinsically non-separable dependencies, MAC-Flow cannot preserve the joint geometry through factorization, even if the centralized flow model is learned perfectly.
>
> Finally, we would like to emphasize our positioning, our solution is designed to retain high expressiveness while providing fast factorized policies at test time. Our goal is not to uniformly outperform diffusion models on every possible coordination, but to offer a practical middle ground. That is, strong coordination performance on complex cooperative tasks with an order-of-magnitude reduction in inference cost. We believe this transparency strengthens the paper by showing when factorization is appropriate, when it is fundamentally limited, and why MAC-Flow offers a compelling trade-off for the majority of offline MARL applications.

---

> ### Author Response · Authors · 2025-11-16
> **Response to reviewer's concerns (5/5)**
>
> **We uploaded the revised manuscript.**
>
> We appreciate the reviewer’s thoughtful questions and the time invested in providing such detailed feedback again. If there are any additional concerns or points that would benefit from further clarification, please feel free to let us know. We would be more than happy to discuss them. Thank you again for your careful evaluation and for helping us strengthen the paper.
>
> ---
> `Reference`
>
> [1] A. Ren, et al. Diffusion policy policy optimization. arXiv 2024.
>
> [2] M. Mark, et al. Policy agnostic RL: Offline RL and online RL fine-tuning of any class and backbone. ICLR 2025 Workshop SSI-FM.
>
> [3] A. Wagenmaker, et al. Steering Your Diffusion Policy with Latent Space Reinforcement Learning. CoRL 2025.

---

### Official Review · Reviewer_4G4R · 2025-10-31

**Soundness:** 3
**Presentation:** 3
**Contribution:** 3
**Rating:** 6
**Confidence:** 2

**Summary:**

This paper tackles a long standing tension in offline multi agent RL between expressive centralized training and fast decentralized execution. The authors first learn a centralized joint policy over all agents’ actions via flow matching, then distill it into one step per agent policies suitable for test time. Empirically, MAC-Flow performs competitively on standard cooperative benchmarks while delivering large gains in inference speed relative to diffusion style generators, and it stays comparable to simpler Gaussian MARL baselines in wall clock.

**Strengths:**

1. Learning a rich joint policy with flow matching, then distilling to per agent policies, directly addresses the coordination vs speed trade off that many of us have run into in offline MARL. I think the training to deployment narrative is easy to follow and feels usable.
2. The comparisons include both diffusion based policies and conventional MARL baselines, and the results highlight a strong reduction in inference latency while retaining returns. Given how often test time latency matters in multi agent control, this is an impactful result.
3. The writing is clear, the algorithmic pipeline is spelled out, and there are helpful ablations that point to which pieces do the heavy lifting.

**Weaknesses:**

1. The reliance on an individual global max style factorization is an obvious pressure point. The paper would be stronger with stress tests on heavily coupled tasks where separability breaks down. Bounds are good, but concrete counterexamples or failure modes would build trust.
2. The link from small $W_2$ to small value loss hinges on a Lipschitz $Q_{tot}$ and on distributional proximity that may not hold uniformly. It would help to see empirical measurements of these quantities during training to show that the assumptions are not only theoretically convenient but also reasonably satisfied.
3. Most of the complexity discussion focuses on test time. Training a flow over joint actions can be expensive, and the main text gives less visibility into training wall clock, memory, and scaling with agent count and action dimension. I want to know the total cost to get the speedups.
4. It is not clear how the approach behaves under partial observability, larger discrete alphabets, or mixed cooperative competitive games. Even a brief study in one of these regimes would help position the method more broadly.

**Questions:**

1. Could you include tasks where interaction strength is dialed up and report both $W_2\big(\pi^{\text{joint}},\prod_i \pi_i\big)$ and the corresponding return gaps?
2. Can you provide training cost curves and memory footprints as functions of agent count and action dimension, along with sensitivity to the number of flow integration steps during training. A head to head with an autoregressive joint policy would be especially informative.
3. How does MAC-Flow adapt when observation is local rather than global? If policies condition on $o_i$ at test time while the joint flow used $s$ and a central critic at training time, do your guarantees change, and what additional assumptions would be needed for the value gap bounds to carry over?

---

> ### Author Response · Authors · 2025-11-16
> **Response to reviewer's concerns (1/4)**
>
> We sincerely thank the reviewer for providing such a thoughtful and deeply engaged evaluation of our work. The reviewer's insights reflect an exceptional level of care and expertise, and they have significantly strengthened the clarity and rigor of our revision. We are truly grateful for the time and depth of analysis you dedicated to this submission!
>
> ---
>
> > **W1 & Q3**  (Failure analysis) Factorization’s bounds are good, but concrete counterexamples or failure modes would build trust.
>
> We thank the reviewer for highlighting the reliance on IGM factorization and the need for concrete counterexamples where separability breaks down. We fully agree that demonstrating explicit failure modes is essential for validating our theoretical assumptions.
>
> `Summary.` We conducted additional experiments to show the failure mode of MAC-Flow and added such results in Section 4.3.
> This toy example illustrates all failure characteristics highlighted by the reviewer: the factorization assumptions provably break, theoretical bounds no longer predict behavior, and MAC-Flow underperforms specifically due to intrinsic non-separability rather than implementation issues. As a result, the XOR stress test offers a controlled and analytically grounded failure mode that directly validates the reviewer’s concerns and strengthens the trustworthiness of our theoretical analysis.
>
> `Update.` In the revision, we added a **Failure Mode Analysis section in Appendix H.6, supported by a stress-test in an analytically tractable XOR coordination environment.** This example is constructed so that the optimal joint action is anti-aligned, making global-max factorization provably impossible, especially the kind of “heavily coupled” setting the reviewer requested.
>
> `Why XOR?` The XOR reward is fundamentally non-separable: the optimal joint action is defined by anti-alignment between the two agents (e.g., $(-1,+1)$ or $(+1,-1)$). Individually, neither agent can determine its optimal action from its own action alone: the reward depends entirely on the other agent’s behavior. Therefore, the individual Q-values collapse to constants, and the per-agent $\operatorname{argmax}$ selects aligned or ambiguous actions (e.g., $(+1,+1)$), which are strictly suboptimal.
>
> Consequently, $\operatorname{argmax}Q_{\text{tot}} \neq (\operatorname{argmax}Q_1, \operatorname{argmax}Q_2)$, making this domain a clean, provably non-IGM-consistent setting.
>
> `Experimental results.`
> - Dataset: The offline data exhibits two sharp, disconnected high-density modes corresponding to the XOR structure.
> - Joint flow policy: Our centralized flow model successfully reconstructs these two modes and captures the non-factorizable geometry of the joint distribution.
> - Factorized policy: When distilled, the per-agent policies collapse to an independent product distribution, entirely missing the high-density anti-aligned regions and instead generating smeared samples near the center.
>
> These distributions are visualized in Figure 14 of Appendix H.6, clearly showing how they break down under strong coupling theoretically. We believe that this experiment provides exactly the type of concrete counterexample the reviewer requested.

---

> ### Author Response · Authors · 2025-11-16
> **Response to reviewer's concerns (2/4)**
>
> > **W2 & Q3**  (Empirical results w.r.t theoretical idea)
>
> We thank the reviewer for raising the concern about whether the theoretical link between small $W_2$ discrepancy and small value loss is empirically justified during training. The reviewer is correct that the guarantee requires (i) a Lipschitz $Q_{\text{tot}}$​ and (ii) sufficiently small distributional mismatch, and these assumptions may not hold uniformly.
>
> To directly address this point, we have run additional experiments using a didactic toy example and updated the results in our revised manuscript. We emphasize that this toy environment is intentionally chosen because it provides a uniquely controlled test bed where the theoretical assumptions can be directly measured and causally validated. This ensures that the mechanism behind the theoretical guarantee is not only mathematically sound but also genuinely reflected in the actual learning dynamics.
>
> `Update.`  In the revision, we added a didactic example in Section 4.3. The measurements in Figure 3 validate that the assumptions underlying our theoretical relation are not only mathematically convenient but are also **reasonably met during training**, thereby strengthening the practical relevance of Proposition 4.3.
>
> `Summary of results.`
>
> - **Value gap decreases in tandem with distillation loss.** In Figure 3(a), the empirical value gap consistently tracks the distillation loss, supporting the theoretical prediction that a smaller discrepancy between joint and factored distributions leads to a proportionally smaller value deviation.
> - **Distributional proximity improves over training.** Figure 3(b) shows that individual policy is mostly lower than joint policy in terms of inter-agent mutual information. Inter-agent mutual information of joint policy steadily decreases, indicating that the learned joint policy distribution clearly captures individual agent's specific action distribution during training. This suggests that the underlying data and task structure admit an increasingly factorable representation over training, making the distributional assumptions in our analysis more plausible in practice.
> - **Empirical value gap remains below the Lipschitz bound.** Figure 3(c–d) shows that all checkpoints lie **strictly below the theoretical Lipschitz envelope** $L_QW_2$​, with a clear monotonic trend between $W_2$​ distances and value gaps. This provides direct empirical evidence that the Lipschitz smoothness of the learned $Q_{\text{tot}}$​ is sufficiently well-behaved in practice.
>
> Taken together, these empirical measurements directly address the reviewer’s request and confirm that the assumptions behind Proposition 4.3 are not only theoretically convenient but indeed observed in practice.

---

> ### Author Response · Authors · 2025-11-16
> **Response to reviewer's concerns (3/4)**
>
> > **W3 & Q2-1**  (Training time) I want to know the total cost of MAC-Flow to get the speed ups. And
>
> We appreciate the reviewer’s concern regarding the training cost of the proposed solution. To address this, we would like to highlight that our original manuscript already provided wall-clock training times for all discrete- and continuous-control benchmarks, summarized in Figure 10 and Appendix H.2.
>
> `Overview.` To directly answer the reviewer’s question, the overall computational cost required to obtain MAC-Flow’s speedup is modest, and our measurements show that MAC-Flow achieves its fast inference without incurring a large training-time overhead. Please refer the our Big O analysis in Appendix C!
>
> `Quick summary of original manuscript.` MAC-Flow trains substantially faster than diffusion-based baselines. In SMAC, Diffusion solutions require about 60 hours, while MAC-Flow trains in 1–5 hours, nearly an order of magnitude faster. In MA-MuJoCo, MAC-Flow completes training on the same datasets in 40–100 minutes, while still outperforming other offline baselines. This total cost is comparable with OMIGA, ICQ, and TD3BC and cheaper than OMAR and CQL.
>
> Next, as per reviewer’s request, we provide a direct comparison of wall-clock training time across the different number of agents as follows.
>
> | Method  | 3m     | 2s3z     | 5m_vs_6m | 8m      |
> | - | - | - | - | - |
> | MABCQ   | 1h 8m   | 1h 44m   | 1h 39m   | 2h 11m  |
> | DoF     | 48h     | 53h      | 54h      | 60h     |
> | MACFlow | 1h 38m  | 2h 46m   | 2h 45m   | 3h 34m  |
>
> This table confirms that MAC-Flow scales linearly with the number of agents. This scaling trend is substantially better than diffusion-based baselines. Compared with MABCQ, MAC-Flow maintains comparable training speed while achieving stronger performance. Overall, these results show that MAC-Flow preserves practical training cost even in larger multi-agent systems, which can also be directly confirmed from the timing table above.
>
> ---
> > Q2-2. Sensitivity to the number of flow integration steps
>
> Thank you for raising this question. We fully acknowledge that flow- and diffusion steps are important hyperparameters. Therefore, we conducted an explicit sensitivity analysis to measure how  the number of flow integration steps affects the performance of MAC-Flow. The results are summarized below (over four seeds):
>
> | flow step |  1 | 4 | 10 | 20 |
> | - | - | - | - | - |
> | 3m | $15.8 \pm 3.6$ | $19.5 \pm 0.5$ | $19.8\pm0.2$ | $19.8 \pm 0.2$ |
> | 6halfcheetah | $3266.5 \pm 1320.7$ | $4302.7 \pm 494.4$ | $4650.0 \pm 271.6$ | $4642.8 \pm 210.6$ |
>
> This table shows that performance of the proposed solution improves rapidly as we increase the number of steps from $1 \rightarrow 4 \rightarrow 10$. However, beyond $10$ steps, the improvement saturates, and the performance with $10$ and $20$ steps is nearly identical on both 3m and 6halfcheetah.
>
> Finally, we would like to highlight that such a trend is consistent with prior findings in the flow-matching literature - flow-based solutions generally exhibit low sensitivity to the number of integration steps, unlike diffusion models [1-5]. Empirically, performance is already stable around 10 steps, and increasing the step sizes only marginal gains.
>
> ---
> > Q2-3. MADT [6] (autoregressive solution) vs MAC-Flow.
>
> Thank you for raising this comparison request. To address the reviewer’s concern, we report the performance of MADT, an autoregressive transformer-based multi-agent policy, as follows. This numerical report comes from DoF paper.
>
> | 3m | Good | Medium | Poor |
> |-|-|-|-|
> | MADT | $19.0 \pm 0.3$ | $15.8 \pm 0.5$ | $4.2 \pm 0.1$ |
> | MAC-Flow |$19.8 \pm 0.2 $ | $18.0 \pm 3.2$ | $10.6 \pm 2.2$ ||
>
> | 8m | Good | Medium | Poor |
> |-|-|-|-|
> | MADT | $18.5 \pm 0.4$ | $18.2 \pm 0.1$ | $4.8 \pm 0.1$  |
> | MAC-Flow | $19.7 \pm 0.3$ | $19.4 \pm 0.6$ | $11.5 \pm 0.8$ |
>
> |3s2z| Good | Medium | Poor |
> |-|-|-|-|
> | MADT | $16.8 \pm 0.1$ | $16.1 \pm 0.2$ | $7.6 \pm 0.3$ |
> | MAC-Flow | $19.5 \pm 0.5$ | $17.6 \pm 0.6$ | $8.5 \pm 0.6$ |
>
> MAC-Flow proves more robust than MADT across all dataset qualities. We conjecture that this performance gap arises because MADT is based on the BC autoregressive model, whereas our solution combines Q-maximization with a high-capacity flow model. Therefore, MAC-Flow outperforms MADT by capturing the joint action distribution and steering it toward high Q regions.
>
> ---
> `Reference.`
>
> [1] C. Durkan, et al. Neural Spline Flows. NeurIPS 2019.
>
> [2] A. Lugmayr, et al. SRFlow: Learning the Super-Resolution Space with Normalizing Flow. ECCV 2020.
>
> [3] C. Lee, et al. FlowCLAS: Enhancing Normalizing Flow Via Contrastive Learning For Anomaly Segmentation. CoRR 2024.
>
> [4] S. Park, et al. Flow Q-learning. ICML 2025.
>
> [5] B. Agrawall, et al. floq: Training Critics via Flow‑Matching for Scaling Compute in Value‑Based RL. arXiv 2025.
>
> [6] Meng, Linghui, et al. Offline pre-trained multi-agent decision transformer. Machine Intelligence Research 20.2 2023.

---

> ### Author Response · Authors · 2025-11-16
> **Response to reviewer's concerns (4/4)**
>
> > **W4** (Scalability) It is not clear how the approach behaves under partial observability, larger discrete alphabets, or mixed cooperative competitive games.
>
> Thank you for your constructive feedback. We fully acknowledge that partial observability, larger discrete alphabets, and mixed cooperative competitive games are significant problems in MARL. Nevertheless, we would like to emphasize that selected benchmarks already span several of the regimes pointed out by the reviewer. The current offline MARL community provides only a limited set of publicly available datasets, and our study covers almost all offline MARL benchmarks that include publicly available datasets.
>
> More precisely, **SMAC** exposes agents to local field-of-view partial observability and strong inter-agent coupling. It is also cooperative-competitive dynamics: although our training datasets do not cover opponent policies directly, the learned team policies must coordinate internally while competing against an adversarial team. **MA-MuJoCo** complements this with a continuous control domain and smooth coordination dynamics by heterogeneous modular agents. **MPE** specifically offers a partially coordinated cooperative scenario, where agents operate under partial observability and must coordinate to cover distinct goals while avoiding collisions.
>
> Additionally, we would like to emphasize the larger discrete alphabet of SMAC, where each agent makes a decision under a 16-dimensional action space. Simply, the flow joint policy requires capturing the distribution of 16 * n (the number of agents). Given we cover a minimum of 3 agents (3m) up to 10 agents (terran_10_vs_10), we consider a maximum of 160 action dimensional space.
>
> While these environments partially cover the axes raised in W4, we fully acknowledge that the current benchmark landscape is still narrow and does not yet offer comprehensive offline datasets for more challenging mixed or partially observable regimes. We appreciate the reviewer’s suggestion, and we plan to extend MAC-Flow by (i) contributing additional offline MARL datasets, and (ii) evaluating richer mixed cooperative-competitive tasks as such datasets become available.
>
> ---
> > **Q3.** Could you include tasks where interaction strength is dialed up and report both $W_2$  and the corresponding return gaps?
>
> Thank you for the insightful suggestion. We fully agree that examining how the algorithm behaves as interaction strength increases is important for validating practical robustness. To answer this question, we have added a new experiment in Appendix H.7 that explicitly dials up the interaction strength and reports both the Wasserstein distance $W_2$​ and the corresponding return gaps.
>
> `Experimental setup (How do we control interaction strength).` We construct a controlled payoff game where the interaction strength $\zeta \in \{0.00, 0.25, 0.50, 0.75, 1.00\}$ is varied by mixing diagonal and off-diagonal joint action modes. In this example, the MDP setup is as follows: each agent can select an action in $\{0, 1\}$, and the team reward is defined as $(0,0) = 0, (1,0)$ or $(0,1) = 1, (1, 1) = 2$.
>
> We define \textit{interaction strength} as the degree to which one agent's optimal action depends on the other agent's action, or equivalently, how far the joint action distribution deviates from a factorized form $p(a^1, a^2) = p(a^1)p(a^2)$. Under this view, diagonal joint actions $(0,0)$ and $(1,1)$ represent strong interaction, because the optimal behavior requires perfectly matching actions and thus induces full coupling. In contrast, off-diagonal actions $(1,0)$ and $(0,1)$ correspond to weak interaction, where each agent's marginal behavior remains nearly independent. To probe this, we generate datasets by mixing diagonal and off-diagonal modes with a controllable interaction parameter $\zeta \in [0,1]$.
>
> `Empirical summary.`
> Return gaps grow with interaction strength. Figure 16(b) shows that the joint policy consistently matches the dataset reward (approximately 1.0), whereas the individual policy returns gradually increase closer to the optimal reward.
> $W_2$ discrepancy also grows monotonically with interaction strength. Figure 16(c) shows that the Wasserstein distance increases steadily as interaction strength increases. This reflects that the joint distribution naively follows dataset, while the individual policy shows biased distribution by structurally following reward signal.
>
> ---
> **We uploaded the revised manuscript.** If there are any additional questions or points that require clarification, we would be more than happy to address them. Thank you again for your thoughtful and constructive feedback.

---

### Official Review · Reviewer_u2dH · 2025-11-01

**Soundness:** 3
**Presentation:** 3
**Contribution:** 3
**Rating:** 6
**Confidence:** 3

**Summary:**

This paper introduces MAC-Flow, a framework for multi-agent coordination that learns a flow-based joint policy from offline data and distills it into decentralized one-step policies to balance performance and inference speed. It achieves ~14.5× faster inference than diffusion-based MARL methods across 4 benchmarks while maintaining good performance, with ablation studies confirming the value of its two-stage strategy.

**Strengths:**

1. MAC-Flow effectively balances the trade-off between multi-agent coordination performance and inference speed.
2. The resulting individual policy supports seamless offline-to-online fine-tuning.
3. The ablation studies show the components of MAC-Flow are effective.

**Weaknesses:**

1. There's still a small performance gap compared to diffusion policies (DoF), especially on SMACv2, showing room for improvement in handling highly stochastic multi-agent environments.
2. It requires offline datasets with diverse joint behaviors for effective training, and its performance may degrade when using low-quality or limited offline data.
3. The baselines for continuous control are weaker due to the absence of diffusion and flow-based policies.

**Questions:**

1. Can you provide some explanations on why MAC-Flow generally underperforms compared to DoF on SMACv2? Does it mean that individual policies are unreliable when facing complex and stochastic environments?
2. How is the training cost of MAC-Flow compared to other baselines?

---

> ### Author Response · Authors · 2025-11-16
> **Response to reviewer's concerns (1/2)**
>
> We sincerely appreciate the time and effort the reviewer dedicated to reviewing our work. This evaluation and constructive feedback are extremely valuable to us. Thank you for helping us further refine and strengthen the paper.
>
> ---
> > **W1 & Q1** SMACv2 performance between MAC-Flow and DoF
>
> Thank you for providing insightful questions and pointing out important points for our future work. There is still a small performance gap compared to diffusion-based policies (DoF), particularly on SMACv2, which reflects that highly stochastic multi-agent environments continue to pose challenges. While this observation is correct, we would like to emphasize that the main claim of our work is to alleviate the fundamental trade-off between coordination quality and inference efficiency, rather than to surpass diffusion-based policies. This claim is already acknowledged by the reviewer (also, two other reviewers) in their strength: the merit of our inference-time speed and performance trade-off. Meanwhile, we agree that the next step is bridging to develop more robust coordination algorithms with MAC-Flow's efficiency.
>
> Importantly, the key contribution of our method lies in the two-stage flow distillation and IGM-based value decomposition, not in any specific backbone. Despite using this simple architecture, MAC-Flow still achieves competitive performance, indicating that the underlying framework is expressive and can naturally accommodate more powerful architectures in future extensions if desired.
>
> **Comparison with low step of diffusion baseline.** We would like to share performance of the small number of diffusion-step on the SMAC v1 3m task to clarify the role of iterative sampling (50 and 100 is reported by DoF paper). As shown below, both baselines degrade substantially when the number of diffusion steps is reduced:
>
> | diffusion step | MADiff | DoF |
> | -- | -- | -- |
> | 50 |13.5 | 12.7 |
> | 100 |16.4| 15.1|
> | 200 |19.8 |19.3 |
>
> These results reflect a general property of diffusion-style policies. Their performance depends on many iterative refinement steps. Our goal is to highlight that MAC-Flow offers a complementary trade-off: single-step action generation with competitive performance, without relying on long denoising trajectories.
>
> Looking forward, we see several promising directions for improving robustness under stochastic dynamics. For example, augmenting the flow model with test time corrective refinement using value gradients, which could provide more stable adaptation under uncertainty.
>
> ---
> > **W2** (General challenge of offline RL) Depends on quality of dataset
>
> We agree with the reviewer that the performance depends on the quality and coverage of the offline dataset. This is a fundamental challenge of offline RL, not unique to our method, and poorly explored joint-behavior data can indeed lead to suboptimal policies. However, MAC-Flow mitigates this issue more effectively than BC-style generative approaches such as MADiff. MAC-Flow incorporates IGM-based value decomposition and explicit Q-maximization, enabling the model to leverage value structure even when behavior data is imperfect, thereby reducing sensitivity to dataset bias.
>
> Furthermore, MAC-Flow offers extensibility that diffusion-based offline baselines lack. As demonstrated in RQ3 (Fig. 4), our method can **be directly adapted for online fine-tuning**. This allows the agent to **quickly correct for dataset deficiencies once limited interaction** becomes available. Note that DoF, MADiff, and other heavy generative models are significantly less suited for.
>
> In summary, while we acknowledge the reviewer’s insight that dataset quality is an intrinsic bottleneck of offline RL, we would like to emphasize that MAC-Flow alleviates this challenge more effectively than prior generative offline MARL methods. Importantly, our empirical results show that this robustness is not limited to high-quality datasets: across SMAC and MA-MuJoCo, MAC-Flow consistently performs strongly under medium, medium-replay, and medium-expert mixtures, where behavior quality is suboptimal.

---

> ### Author Response · Authors · 2025-11-16
> **Response to reviewer's concerns (2/2)**
>
> > **W3** (Baseline comparison) The absence of diffusion baseline in continuous control
>
> We acknowledge the reviewer’s concern regarding the absence of diffusion-based baselines in the continuous-control setting. To address this, we have run additional experiments over six seeds in MA-MuJoCo, and we have reported the MPE score, which is provided by the MADiff paper.
>
> | MPE | Expert | Medium | Medium-Replay | Random |
> | - | - | - | - | - |
> | MADiff | $95.0\pm5.3$ | $64.9\pm7.7$ | $30.3\pm2.5$ | $6.9\pm3.1$ |
> | MAC-Flow | $101.7\pm10.9$ | $80.1\pm20.6$ | $50.4\pm33.2$ | $31.1\pm6.8$ |
>
> | 2ant | Expert | Medium | Medium-Replay | Medium-Expert |
> | - | - | - | - | - |
> | MADiff | $2060.0 \pm 10.3$ | $1428.4 \pm 14.7$ | $1294.5 \pm 360.2$ | $1740.2 \pm 158.9$ |
> | MAC-Flow | $2060.2 \pm 20.0$ | $1432.4 \pm 17.8$ | $1498.4 \pm 20.3$ | $2053.3 \pm 20.4$ |
>
> | 3hopper | Expert              | Medium              | Medium-Replay       | Medium-Expert       |
> | - | - | - | - | - |
> | MADiff              | $2853.3 \pm 593.8$  | $1436.8 \pm 449.5$  | $936.1 \pm 574.0$    | $2810.4 \pm 723.2$   |
> | MAC-Flow            | $3592.1 \pm 8.9$    | $1023.5 \pm 253.0$  | $1166.3 \pm 451.9$   | $2988.3 \pm 480.2$   |
>
> | 6halfcheetah | Expert               | Medium               | Medium-Replay        | Medium-Expert        |
> | - | - | - | - | - |
> | MADiff                   | $4711.4 \pm 213.6$   | $2650.0 \pm 365.4$   | $2830.5 \pm 292.8$     | $4410.9 \pm 836.8$     |
> | MAC-Flow                 | $4650.0 \pm 271.6$   | $4358.5 \pm 369.2$   | $3030.2 \pm 436.8$     | $5139.9 \pm 84.1$      |
>
> These results show that MAC-Flow performs competitively despite using a substantially simpler backbone and a single-step policy structure. Importantly, our goal is not to surpass diffusion models in absolute expressiveness, but to **alleviate the core trade-off between coordination quality and inference efficiency**. The continuous-control experiments further support this: MAC-Flow achieves strong performance without relying on iterative denoising, offering an efficient and scalable alternative when fast inference is needed.
>
> ---
> > **Q2** (Training time) Training cost of MAC-Flow
>
> We appreciate the reviewer’s concern regarding the training cost of the proposed solution. To address this, we would like to highlight that our original manuscript already provided wall-clock training times for all discrete- and continuous-control benchmarks, summarized in Figure 10 and Appendix H.2. However, we agree that this discussion is important for understanding the overall computational trade-off of MAC-Flow, and we included a brief summary of the training time results in the main body,  with full details in Appendix H.2.
>
> `Quick summary of original manuscript` MAC-Flow trains substantially faster than diffusion-based baselines. In SMAC, Diffusion solutions require about 60 hours, while MAC-Flow trains in 1–5 hours, nearly an order of magnitude faster. In MA-MuJoCo, MAC-Flow completes training on the same datasets in 40–100 minutes, while still outperforming other offline baselines. This total cost is comparable with OMIGA, ICQ, and TD3BC and cheaper than OMAR and CQL.
>
> Furthermore, we provide a direct comparison of wall-clock training time across the different number of agents as follows.
>
> | Method  | 3m     | 2s3z     | 5m_vs_6m | 8m      |
> | - | - | - | - | - |
> | MABCQ   | 1h 8m   | 1h 44m   | 1h 39m   | 2h 11m  |
> | DoF     | 48h     | 53h      | 54h      | 60h     |
> | MACFlow | 1h 38m  | 2h 46m   | 2h 45m   | 3h 34m  |
>
> This table confirms that MAC-Flow scales linearly with the number of agents. This scaling trend is substantially better than diffusion-based baselines. Compared with MABCQ, MAC-Flow maintains comparable training speed while achieving stronger performance. Overall, these results show that MAC-Flow preserves practical training cost even in larger multi-agent systems, which can also be directly confirmed from the timing table above.
>
> ---
> Should you have any additional questions or require further clarification, we would be pleased to provide it. Thank you once again for your insightful and constructive comments.

---

### Author Response · Authors · 2025-11-19
**Global Comment for paper revision**

Dear AC and all reviewers,

We express our gratitude to AC for managing the review process and all three reviewers for their insightful feedback.  We are pleased to present the updates we have made in response to valuable suggestions, as detailed below.

* Added a clarification sentence explaining the assumption and theoretical bound (Line 317) to improve conceptual transparency
* Included a didactic example that offers empirical evidence supporting the proposed mathematical formulation
* Added additional clarification regarding our focus when comparing against prior solutions (Line 430) to avoid potential misunderstandings
* Introduced new experimental results for MADiff (diffusion-based baseline) in continuous control settings, now reported in Table 2
* Added scalability experiments evaluating performance as the number of agents increases, provided in Appendix H.4 with corresponding Figure 13
* Added ablation experiments analyzing MAC-Flow’s advantages compared to a naive BC generative modeling and a variant without IGM, included in Appendix H.5 and Figure 14
* Introduced failure mode analysis experiments for MAC-Flow, presented in Appendix H.6 and Figure 15
* Added interaction-strength analysis experiments to study coordination sensitivity, provided in Appendix H.7 and Figure 16


We have left individual comments to address each reviewer’s questions, concerns, and our misclarification. Please review each individual comment for more details!

---

> ### Author Response · Authors · 2025-12-01
> **Final summarization of author-reviewer discussion**
>
> Dear AC,
>
> We would like to provide you with a brief summary of the rebuttal context. Although the discussion period unfortunately did not involve much interaction with the reviewers, their initial comments were constructive, and we have made our best effort to address them in our responses.
>
> ### **TL;DR**
> This paper introduces MAC-Flow, which learns an expressive flow-based joint policy and distills it into fast decentralized one-step policies, alleviating the performance-inference speed trade-offs. It achieves $\sim 14.5\times$ faster inference than diffusion-based MARL methods while maintaining comparable coordination performance.
>
> ### **Strengths**
>
> - Directly **tackles the long-standing performance–inference speed trade-off**, delivering diffusion-level coordination with Gaussian-level efficiency `u2dH` `4G4R` `Z5Fb`
> - Clearly states the technical solution and the previous trade-off problem. `4G4R` `Z5Fb`
> - Include **helpful ablation** for explaining algorithmic design choices. `u2dH` `4G4R`
> - Presents good properties of MAC-Flow.
>     - It supports **seamless offline-to-online fine-tuning**. `u2dH`
>     - The deployment narrative is **easy** to follow and feels **usable**. `4G4R`
>     - It **provides a standard** with theoretical IGM for **scalable** deployment.  `Z5Fb`
>
> ### **Raised concerns and our responses/revisions**
> - **Baseline comparison.**
>     - We conducted additional experiments and confirmed that our solution is still comparable with MADIFF in the continuous action domain. `u2dH`
>     - We compared our solution with the autoregressive solution and confirmed that our solution outperforms MADT. `4G4R`
> - **Stress test and empirical evidence** for MAC-Flow's theoretical idea. `4G4R` `Z5Fb`
>     - (**Added Section 4.3**) We provided a didactic example for $W_2$-value gap validation.
>     - (**Added Appendix H.5**) We provided a pure coordination game for explaining why Mac-Flow performs better than BC generative modeling and why we consider IGM.
>     - (**Added Appendix H.6**) We provided a failure mode analysis for MAC-Flow’s transparency.
>
> ### **Discussion points**
> - **Partial observability and scalability**. `4G4R` `Z5Fb`
>     - (**Added Appendix H.5**) We added a landmark-covering game to systematically analyze scalability, evaluating MAC-Flow as the number of agents increases from 3 up to 40.
>     - All benchmarks we use are already partially observable settings.
> - **Behavioral properties** of MAC-Flow according to level of **interaction strength**.  `4G4R`
>     - (**Added Appendix H.7**) We added a payoff game for analysis according to interaction strength
> - **Training-time overhead** analysis. `u2dH` `4G4R`
>     - Our original manuscript already includes training time analysis in Appendix H.2. MAC-Flow achieves fast inference at the cost of a modest training-time overhead.
> - **Small performance gap** compared to diffusion policies. `u2dH`
>     - Our goal is not to surpass diffusion models in absolute expressiveness, but to alleviate the core performance-inference speed trade-off.
>
> We are genuinely thankful for the time and attention that the AC has devoted to ensuring a fair and balanced evaluation. We would like to emphasize that, thanks to the constructive feedback from all reviewers, we conducted a broad set of additional analyses in this rebuttal. Since these improvements are difficult to summarize numerically, we kindly ask the AC to review the newly added sections.

---

### Meta-Review · Area_Chair_odX1 · 2025-12-16

**Summary:**

This submission proposes MAC-Flow, an offline MARL framework that learns a flow-matching joint policy from offline data and distills it into decentralized one-step per-agent policies guided by IGM-based value maximization. The approach targets the coordination quality–inference latency trade-off and offers a clean, implementable pipeline, further strengthened in the rebuttal by additional analyses on assumptions, failure modes, scalability, and baseline coverage.

The First two reviewers already provided positive scores, emphasizing the practical relevance and the strong empirical results. Reviewer Z5Fb’s concerns were largely mitigated by the revision, which clarifies the theoretical claim as a bounded degradation or value-gap characterization rather than a global-optimality preservation theorem, and adds scaling evidence to support robustness with larger agent populations. While some items remain partially unaddressed, such as fully reproducible head-to-head comparison with a specific recent diffusion-based method and broader stress tests, these do not undermine the core contribution. Taking the overall consensus, the strengthened rigor, and the clear practical utility together, I recommend acceptance.

**Reviewer Concerns:**

1. Head-to-head comparison with “Graph Diffusion for Robust Multi-Agent Coordination (2025)” requested by Z5Fb: the authors plan to cite/discuss it but state a reproducible empirical comparison is infeasible due to unavailable code and differing protocols; thus, this remains partially unresolved as a strict baseline check

**Reviewer Scores:**

Reviewer u2dH:  6.
Reviewer 4G4R:  6.
Reviewer Z5Fb: 4.

---

### Decision · Program_Chairs · 2026-01-26

Accept (Poster)